# SPP1 + macrophages cause exhaustion of tumor-specific T cells in liver metastases

Rajiv Trehan[1], Patrick Huang[1,6], Xiao Bin Zhu[1,6], Xin Wang [1,6], Marlaine Soliman[1], Dillon Strepay [2], Amran Nur[1], Noemi Kedei[3], Martin Arhin [4], Shadin Ghabra [1], Francisco Rodríguez-Matos [1], Mohamed-Reda Benmebarek [1], Chi Ma [1], Firouzeh Korangy [1,7] & Tim F. Greten [1,5,7] ✉

Functional tumor-specific CD8+ T cells are essential for effective anti-tumor immune response and immune checkpoint inhibitor therapy. Here we show that, compared to other organ sites, primary, metastatic liver tumors in murine models contain a higher number of tumor-specific CD8+ T cells which are also dysfunctional. High-dimensional, multi-omic analysis of patient samples reveals a higher frequency of exhausted tumor-reactive CD8+ T cells and enriched interactions between these cells and SPP1+ macrophages in profibrotic, alpha-SMA rich regions specifically in the liver. Differential pseudotime trajectory inference analysis reveals that extrahepatic signaling promotes an intermediate cell (IC) population in the liver, characterized by co-expression of VISG4, CSF1R, CD163, TGF-βR, IL-6R, and SPP1. Analysis of premetastatic adenocarcinoma patient samples reveals enrichment of this population may predict liver metastasis. These findings suggest a mechanism by which extrahepatic tumors drive liver metastasis by promoting an IC population that inhibits tumor-reactive CD8+ T cell function.

The liver is one of the most common sites for cancer metastasis. Up to 50% of cancer patients will present with or develop liver metastases during their course of disease[1]. Unfortunately, the prognosis of patients with liver cancer metastasis is poor even after systemic therapy and resection[1]. Moreover, patients with liver metastases receiving immune checkpoint inhibitors have significantly worse overall and progression-free survival than those without liver metastasis[2]. This may be due to the liver's unique immune environment, favoring immune tolerance, which affects immunotherapy responses[2]. Immune checkpoint inhibitor therapy aims to reinvigorate tumor-reactive CD8 + T cells, and new mechanistic insights drive novel treatment options[3].

Tumor-reactive CD8+ T cells mediate anti-tumor immunity upon checkpoint inhibitor therapy and vaccines[4,5]. Various mechanisms leading to CD8+ T cell dysfunction have been described. The mechanisms behind tumor-reactive CD8+ T cell dysfunction have been studied since the defining in 1968 of the Hellström paradox - the highly immunosuppressive tumor microenvironment (TME) must be overcome for effective immunotherapy[4]. Similar to peripheral tolerance, tumor neoantigens early in tumor progression induce a hyporesponsive CD8 + T cell state[4]. During late tumor progression, persistent antigen stimulation and the induced presence of immunosuppressive cells cause tumor-reactive CD8 + T cell dysfunction[4].

[1]Gastrointestinal Malignancy Section, Thoracic and Gastrointestinal Malignancies Branch, Center for Cancer Research, National Cancer Institute, National Institutes of Health, Bethesda, MD, USA. [2]Auditory Development and Restoration Program, National Institute on Deafness and Other Communication Disorders, National Institutes of Health, Bethesda, MD, USA. [3]Collaborative Protein Technology Resource, OSTR, Office of the Director, Center for Cancer Research, National Cancer Institute, National Institutes of Health, Bethesda, MD, USA. [4]Neurosurgery Unit for Pituitary and Inheritable Diseases, National Institute of Neurological Diseases and Stroke, National Institutes of Health, Bethesda, MD, USA. [5]NCI CCR Liver Cancer Program, National Institutes of Health, Bethesda, MD, USA. [6]These authors contributed equally: Patrick Huang, Xiao Bin Zhu, Xin Wang. [7]These authors jointly supervised this work: Firouzeh Korangy, Tim F. Greten. ✉e-mail: tim.greten@nih.gov

Numerous approaches for defining potentially tumor-reactive CD8 + T cells (pTRT) in single-cell RNA-sequencing (scRNA-seq) data exist[6–12]. Each approach tries to address the challenges of nonspecific classification of bystander T cells in tumors as well as the over-representation of exhausted pTRT clusters[6–12]. Bystander T cell populations have been overrepresented despite the use of TCR clonality, application of TCR signaling signatures, and utilization of limited markers (CD39, PD1, Lag3, TCF7)[6–12]. Recent publications attempting to address these issues have instead used extensive gene signatures to profile tumor-reactive T cells across multiple tumor types, including colon cancer[13,14].

T cell infiltration has been classically placed under the "hot and cold paradigm," where M2 macrophages play a major role in the immunosuppressive microenvironment of cold tumors[15]. M2 tumor-associated macrophages (TAMs) have been associated with resistance to immunotherapy and have been a target for drug development[15]. In contrast, bioinformatic analysis has postulated SPP1 macrophages to be distinct from the conventional M1 or M2 dogma, highlighting the need for in vivo data[16]. However, there remains a lack of mechanistic insight into their specific role and function within the TME[16].

Here, we apply a high-dimensional, integrated approach utilizing three human scRNA-seq data sets (with appropriate per-patient analysis), proteomic CODEX data, ex vivo systems, and murine in vivo models to identify an M2-independent, targetable mechanism of tumor-reactive CD8 + T cell dysfunction specific to the liver. In addition, we find that this mechanism provides a paradigm for cancer cell metastasis to the liver through a unique IC population affecting pro-fibrotic polarization.

## Results

### A paradoxical high frequency of tumor-antigen-specific (TAS) CD8 + T cells in tumor-bearing livers

The liver is a common site for metastasis from colon cancer. The frequency and phenotype of TAS CD8 + T cells were studied in mice bearing CT26 colorectal carcinoma using the well-established MHC class I tetramer loaded with AH1 peptide[17,18]. CT26 cells were implanted into the liver to form intrahepatic (IH) tumors and into the skin to form subcutaneous (SQ) tumors. First, we investigated the kinetics of TAS CD8 + T cells 7, 14, and 21 days post-tumor injection in mice with intrahepatic or subcutaneous CT26 tumors. The frequencies of TAS CD8 + T cells in the tumor, spleen, and tumor-free liver tissues were measured. Time course analysis revealed a marked early increase of TAS CD8+ T cells (~ 10%) in intrahepatic tumor-infiltrating lymphocytes (TIL) one week after tumor injection compared to minimal levels of TAS CD8+ T cells in other tested tissues. The relatively higher frequency of TAS CD8 + T cells in intrahepatic TILs compared to subcutaneous TILs persisted throughout the entire observation period, peaking on day 14 (Fig. 1A). The number of TAS CD8 + T cells increased over time in intrahepatic tumor-infiltrating lymphocytes (TIL) (Fig. 1A). On day 14, the frequency of TAS CD8+ T cells was higher in the intrahepatic TILs compared to subcutaneous TILs, whereas no significant differences in tumor weights between intrahepatic and subcutaneous CT26 tumors were found at this time point, ruling out the contribution of tumor burden (Fig. 1B, C). Similar trends were seen when analyzing absolute counts (cells/gram tissue) on days 14 and 21 (Supplementary Fig. 1A–C). Interestingly, the tumor-free liver tissue surrounding the intrahepatic tumors also showed an increase in the frequency of TAS CD8 + T cells over time (Fig. 1A). The frequency of TAS CD8+ T cells was lower in the spleen of mice with hepatic tumors as well as subcutaneous tumors. Both tumor-free mice and subcutaneous tumor-bearing mice showed consistently low frequency of TAS T cells in the liver across all analyzed time points (Fig. 1A). Strikingly, the intrahepatic CT26 tumors grew at a faster rate than their subcutaneous counterparts despite the presence of a high number of tumor-infiltrating TAS CD8 + T cells (Fig. 1B, C)[19].

Next, we included the lung, another common organ to form metastasis. CT26 cells were implanted into the liver to form intrahepatic (IH) tumors, into the skin to form subcutaneous (SQ) tumors, or injected into the tail vein (TV) to form intrapulmonary tumors. Tumor, liver, lung, and spleen tissue were harvested on day 10 for flow cytometric analysis. The highest number of TAS CD8 + T cells (12.45%) was seen in tumor-infiltrating CD8 + T cells in intrahepatic tumors (Fig. 1D, E). The portal lymph node from liver tumor-bearing mice also showed a higher frequency of TAS CD8 + T cells compared to hilar and cutaneous lymph nodes (Supplementary Fig. 1D). A lower frequency (0.455%) of TAS CD8 + T cells was found in the spleen of the intrahepatic model (Supplementary Fig. 1D). In contrast to tumor-bearing lungs, the subcutaneous TILs had a high frequency (3.75%) of TAS CD8 + T cells (Fig. 1D). Furthermore, lower numbers of TAS CD8 + T cells were found in the livers of mice with subcutaneous and pulmonary tumors (Supplementary Figs. 1E and F). To confirm the finding, TAS CD8 + T cells were also tested in the second model with B16F10 melanoma using an MHC class I tetramer loaded with TRP2 peptide, and similar results were found (Supplementary Figs. 1G–I)[17,18]. In summary, tumors growing in the liver resulted in the highest frequency of TAS CD8 + T cells.

### Phenotype and function of TAS CD8 + T cells

To characterize the TAS CD8 + T cells in the intrahepatic and subcutaneous tumors, we characterized the phenotype of TAS CD8 + T cells by flow cytometry using markers of cell exhaustion (CD39, TIM-3, PD-1), cytotoxicity (Granzyme B), activation (CD69, PD-1) and proliferation (Ki67) (Fig. 2, Supplementary Fig. 2)[20–22]. We also compared the AH1 + (TAS) CD8 + T cells to the AH1$_{neg}$ CD8 + T cells, which included the bystander CD8 + T cell population over time (Fig. 2B, D, F, and H).

We found the highest frequency of CD39 + TAS CD8 + T cells over time in the TILs and livers of mice with intrahepatic tumors. On day 14, intrahepatic TILs had a significantly higher frequency of CD39 + TAS T cells compared to subcutaneous TILs, which lasted until day 21. By day 21, the majority (89.7%) of the intrahepatic TILs expressed CD39 (Fig. 2A)[20–22]. Notably, both livers from subcutaneous tumor-bearing mice and spleens from intrahepatic tumor-bearing mice showed an increase in the frequency of CD39 + TAS CD8 + T cells on day 21 (Fig. 2A). AH1 + (TAS) CD8 + T cells showed an approximate four-fold increase in CD39 expression in the liver and a two-fold increase in the TILs of liver tumor-bearing mice compared to AH1$_{neg}$ CD8 + T cells (Fig. 2B). Similar results were found for TIM-3 expression (Supplementary Fig. 3A, B). Next, we studied Gzmb expression on CD8 + T cells.

The expression of the cytotoxic marker Gzmb in TAS CD8 + T cells showed an overall pattern of initial increase followed by a sharp drop following tumor progression. The frequency of Gzmb + TAS CD8 + T cells both in the intrahepatic TILs and surrounding liver tissue, as well as subcutaneous TILs, increased at day 14 but drastically decreased by day 21 (Fig. 2C). On day 14, AH1 + CD8 + T cells expressed increased levels of Gzmb compared to AH1$_{neg}$ cells both in the intrahepatic TILs and the surrounding hepatic tissue of tumor-bearing livers (Fig. 2D).

There was a consistently high frequency (96.84% on day 21) of CD69 + TAS CD8 + T cells over time in the TILs and livers of mice with liver tumors as well as in the TILs of mice with subcutaneous tumors (Fig. 2E). Similar to CD39, both the livers from mice with subcutaneous tumors and the spleens from mice with hepatic tumors showed an unexpected increase in the frequency of CD69+ TAS CD8+ T cells on day 21 (Fig. 2E). This points to an activated phenotype in TAS CD8 + T cells in both the liver and tumor after intrahepatic injection. The frequency of CD69 + AH1 + (TAS) CD8 + T cells was higher than AH1$_{neg}$ CD8 + T cells in both the liver and the TIL of liver tumor-bearing mice (Fig. 2F). Similar results were found for PD-1 (Supplementary Fig. 3C and D)[23].

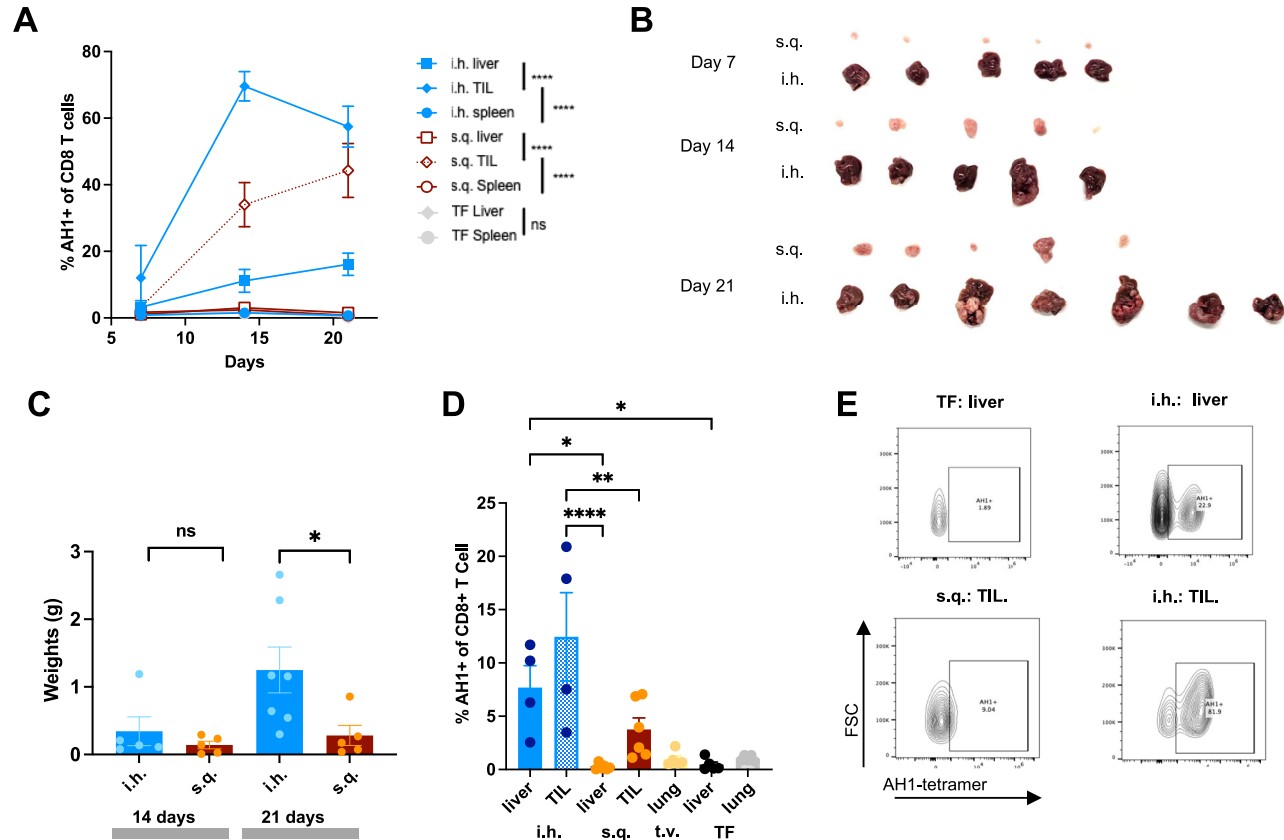

**Fig. 1 | Tumor-Antigen-Specific (TAS) CD8 T Cells Across Murine Models.**
**A** Frequency of AH1 + CD8 + T cells in liver and subcutaneous tumor-bearing mice compared to tumor free (TF) mice at days 7 (subcutaneous (s.q.), $n = 5$; intrahepatic (i.h.), $n = 5$, tumor injection), 14 (s.q., $n = 5$; i.h., $n = 5$) and 21(s.q., $n = 5$; i.h., $n = 7$) (i.h. liver vs i.h. TIL $p < 0.0001$; i.h. Spleen vs i.h. TIL $p < 0.0001$; s.q. liver vs s.q. TIL $p < 0.0001$; s.q. TIL vs s.q. spleen $p < 0.0001$; TF Liver vs TF Spleen $p > 0.9999$).
**B** Photo showing excised subcutaneous and intrahepatic tumors across days 7 (i.h., $n = 5$; s.q., $n = 5$; TF, $n = 5$), 14 (i.h., $n = 5$; s.q., $n = 5$; TF, $n = 5$), and 21 (i.h., $n = 7$; s.q., $n = 5$; TF, $n = 5$). **C** Tumor weights at day 14 ($p = 0.8320$) and 21 ($p = 0.0232$) (i.h., $n = 5$; s.q., $n = 5$;). Mice were euthanized at 3 weeks when tumors had reached

humane clinical endpoints, and, at day 21, the intrahepatic tumors had an average weight of 1.3 g while subcutaneous tumors weighed 0.3 g. **D** Frequency of AH1 + CD8 + T cells in various organs; i.h., $n = 4$; TV, $n = 5$; s.q., $n = 6$; TF, $n = 5$ (i.h. TIL vs s.q. liver $p < 0.0001$; i.h. TIL vs s.q. TIL $p = 0.0047$; i.h. liver vs s.q. liver $p = 0.0212$; i.h. liver vs TF liver $p = 0.0361$). **E** Representative flow cytometry plots of D. All experiments (**A**–**E**) were completed in female, BALB/c 6-8 week old mice. Data was analyzed using 2-way ANOVA for **A** and ordinary one-way ANOVA for **C** and **D**: $*P < 0.05$; $**P < 0.01$; $***P < 0.005$; $****P < 0.001$, ns = not significant (data are presented as mean values +/− SEM).

There was no discernible pattern in the expression of Ki67 in TAS CD8 + T cells over time within the intrahepatic and subcutaneous TILs, indicating that the frequency of TAS CD8+ T cells was not due to an increase in proliferation (Fig. 2G). AH1 + (TAS) CD8 + T cells showed an approximately three-fold higher Ki67 + frequency in the liver with no difference in the TILs of liver tumor-bearing mice compared to AH1$_{neg}$ CD8 + T cells (Fig. 2H).

Next, we evaluated the effect of CD8 + T cell depletion on the growth of intrahepatic and subcutaneous tumors. CD8 + T cell depletion using an anti-CD8α antibody was confirmed by flow cytometry (Supplementary Fig. 3E). Depletion of CD8 T cells did not affect the growth of intrahepatic tumors while subcutaneous tumors were significantly larger (Fig. 2I and Supplementary Fig. 3F). Thus, we conclude that dysfunctional CD8 + T cells in the liver failed to control intrahepatic tumor growth but effectively controlled the growth of subcutaneous tumors.

In summary, these findings indicate that TAS CD8 + T cells showed an exhausted and dysfunctional phenotype in mice with liver tumors. Next, we decided to study human TAS CD8 + T cells within and outside the liver, as well as the interaction patterns of the intrahepatic TAS CD8 + T cells with other cells in the tumor microenvironment.

## SPP1 + macrophages interact with pTRT cells in humans
Given our findings from the mouse studies, we decided to investigate human tumor-reactive CD8 + T cells using publicly available processed and annotated scRNA-seq data from 10 patients with primary colorectal cancer tumor (PT), surrounding colonic tissue (PN), metastatic intrahepatic tumor (MT), surrounding hepatic tissue (MN), tumor-draining lymph nodes (LN), and peripheral blood mononuclear cells (PBMC) (Fig. 3A)[24]. We obtained and validated a previously published cluster annotation for this data[24]. A total of 304,816 cells among 78 clusters with published annotations were obtained, from which 26 represented myeloid clusters (Supplementary Fig. 4A, B)[24]. Additionally, 14 CD8+ T cell clusters (previously published) were obtained including an (1) exhausted cluster expressing LAYN and LAG3 (10: hC11_CD8_Tex-LAYN), (2) tissue-resident memory cluster expressing KLRB1 and KLRD1 (9: hC10_CD8_Trm-KLRB1), (3) CD160 IEL cluster expressing KLRC2 (6: hC07_CD8_IEL-CD160), (4) HSPA1A cluster expressing HSPA6 and HSPA1B (11: hC12_CD8-HSPA1A) and a (5) proliferative cluster expressing MKI67 (13: hC14_CD8_MAIT-SLC4A10) (Supplementary Fig. 4C)[24].

We used single-cell Gene Set Enrichment Analysis (scGSEA) with more than 100 gene signatures to identify potentially tumor-reactive (pTRT) human CD8+ T cells (Fig. 3A)[13,14]. We applied

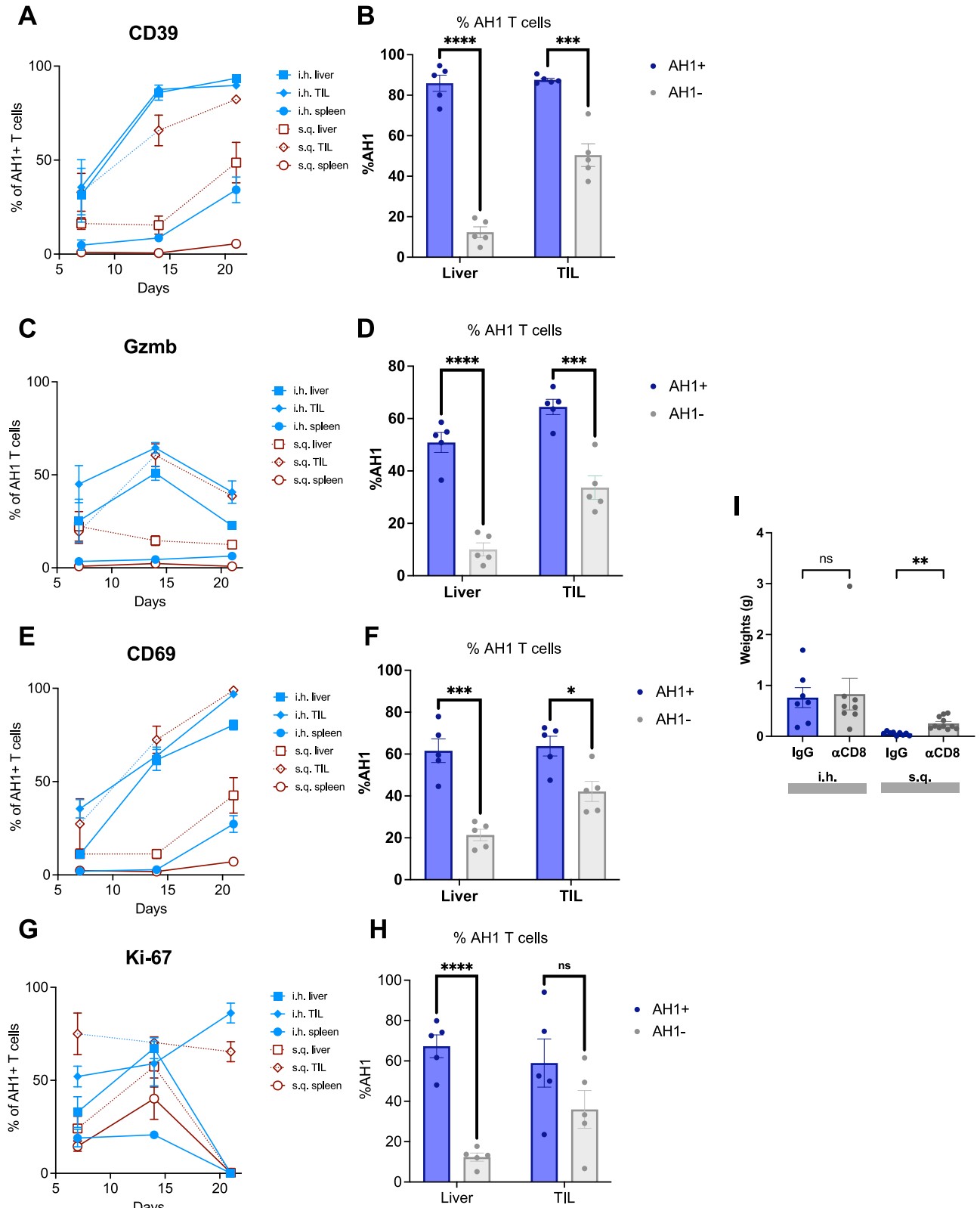

**Fig. 2 | TAS CD8 + T cell phenotype and function.** Frequency of (**A**) CD39 +, (**C**) Granzyme B +, (**E**) CD69+, and (**G**) Ki-67+ among TAS CD8+ T cells in the TIL, liver, and spleen of liver and subcutaneous tumor-bearing mice at days 7 (subcutaneous (s.q.), $n = 5$; intrahepatic (i.h.) tumor injection, $n = 5$), 14 (s.q., $n = 5$; i.h., $n = 5$) and 21(s.q., $n = 5$; i.h., $n = 7$). Comparison of (**B**) CD39+ (Liver $p < 0.000001$; TIL $p = 0.000178$), (**D**) Gzmb+ (Liver $p = 0.000037$; TIL $p = 0.000413$), (**F**) CD69+ (Liver $p = 0.000404$; TIL $p = 0.012408$); and (**H**) Ki-67+ (Liver $p = 0.000032$; TIL $p = 0.168307$) AH1 + to AH1- CD8 + T cells in the liver (left, $n = 17$) and TIL (right,

$n = 17$) at day 14. **I** Tumor weights after either IgG or αCD8 T cell antibody treatment after intrahepatic (IgG, $n = 7$; αCD8, $n = 8$, $p > 0.9999$) or subcutaneous injection (IgG, $n = 9$; αCD8, $n = 11$, $p = 0.0011$) of tumor. All experiments (**A**–**I**) were completed in female, BALB/c 6-8 week old mice. Data was analyzed with unpaired two-sided $t$ tests (**B**, **D**, **F**, **H**) and brown-forsythe and welch anova (I): *$P < 0.05$; **$P < 0.01$; ***$P < 0.005$; ****$P < 0.001$, ns = not significant (data are presented as mean values +/− SEM).

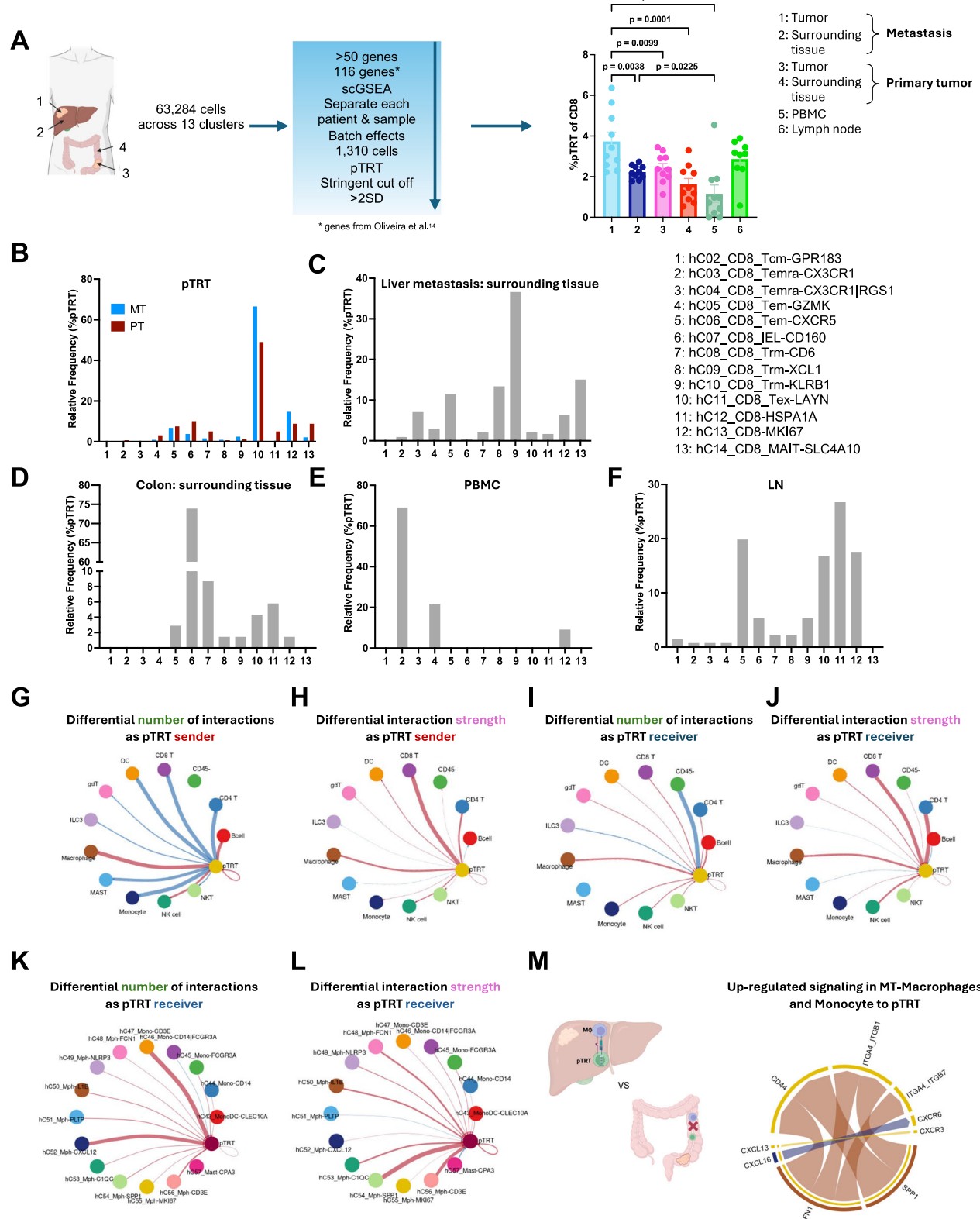

highly stringent cutoffs for the identification of pTRT cells to facilitate the identification of biologically valid mechanisms. CD8 + T cells (1310 cells) that met this cutoff were labeled as pTRT cells, and the rest of these cells were labeled as nonreactive CD8+ T cells hereafter (Supplementary Figs. 4D–F). As a negative control, scGSEA was performed using a randomly generated set of 500 genes, which showed no differences across any organ sites (Supplementary Fig. 5A).

The highest frequency of pTRT CD8 + T cells among CD8+ T cells was found in TILs derived from liver metastasis, followed by lymph nodes and primary CRC TILs (Fig. 3A and Supplementary Figs. 4F, 5B). Moreover, the frequency of intrahepatic pTRT CD8 + TILs was higher than that in the surrounding hepatic tissue (Fig. 3A), analogous to findings in mice (Fig. 1E). We also found a high pTRT frequency in tumor-draining lymph nodes in the human data set (Fig. 3A), analogous to our results in mice (Supplementary Fig. 1D).

**Fig. 3 | scRNA-seq analysis of human tumor-reactive CD8 + T cells. A** Schematic of scRNA-seq analysis to define potentially tumor-reactive (pTRT) CD8 + T cells using single-cell gene set enrichment analysis (scGSEA) and frequency of pTRT CD8 + T cells in various organs; hepatic metastasis, $n = 10$; hepatic surrounding tissue, $n = 10$; primary CRC tumor, $n = 10$; colon surrounding tissue, $n = 10$; PBMC, $n = 10$; lymph node, $n = 9$. Cell-type icons created with BioRender.com[84]. Frequency of pTRT that are found in each CD8 + T cell cluster in (**B**) hepatic metastisis ($n = 10$) compared to primary CRC tumor ($n = 10$). Frequency of pTRT that are found in each CD8 + T cell cluster in (**C**) hepatic surrounding tissue ($n = 10$), (**D**) colon surrounding tissue ($n = 10$), **E**) PBMC ($n = 10$) and (**F**) lymph node ($n=9$) with cluster labels found on the right of (**C**). CellChat cellular communication analysis for communications that the pTRT are sending to all major cell types, comparing hepatic metastasis to primary CRC tumors in the (**G**) number of interactions and (**H**) strength of interactions based on the law of mass action. All interactions that were

found to be statistically greater in the hepatic metastasis ($n = 10$) are shown in red, and in the primary CRC tumor ($n = 10$) are shown in blue. Similarly, CellChat cellular communication analysis for communications that the pTRT are receiving from all major cell types, comparing the (**I**) number of interactions and (**J**) strength of interactions between hepatic metastasis to primary CRC tumors. CellChat cellular communication analysis for communications that the pTRT are sending to macrophage clusters, comparing hepatic metastasis to primary CRC tumors in the (**K**) number of interactions and (**L**) strength of interactions. **M** Schematic of CellChat pathway analysis comparing macrophage-pTRT CD8 + T cell interaction pathways upregulated in hepatic metastasis ($n = 10$) compared to primary CRC tumor ($n = 10$). Cell-type icons created with BioRender.com[84]. Data were analyzed using paired one-way anova analysis (mixed effect analysis) (**A**) and a paired Wilcox test (default for CellChat) (**G**–**M**) (data are presented as mean values +/− SEM).

We then determined the distribution of pTRT cells in previously identified T cell clusters for each tissue type (Fig. 3B–F). Interestingly, pTRT cells were identified across a range of T cell clusters, which allowed for the study of both non-exhausted and exhausted pTRT cells. The majority (66.58%) of pTRT in intrahepatic TILs were found in the exhausted cluster (10: hC11_CD8_Tex-LAYN) (Fig. 3B), similar to our murine models (Fig. 2A, D). A greater proportion of the pTRT cells resided in the T exhausted cluster (LAYN) in intrahepatic TILs compared to primary CRC TILs (Fig. 3B). The majority of pTRT in colonic tissue were CD160+ intraepithelial lymphocytes (6: hC07_CD8_IEL-CD160), while most of the pTRT in the surrounding hepatic tissue were tissue-resident memory (TRM) cells (9: hC10_CD8_Trm-KLRB1) (Fig. 3C, D). CD160 + IEL have been reported to show a cytotoxic function, particularly in the setting of viral diseases. In addition, TRM T cells have been correlated to improved response to immunotherapy as these cells can be locally activated to provide anti-tumor function[25–27]. The majority of pTRT cells in the lymph nodes were HSPA1A + (11: hC12_CD8-HSPA1A) which is enriched in stressed T cells correlating to clinically nonresponsive tumors (Fig. 3F)[28]. The lymph node pTRT cells also highly expressed the proliferative marker MKI67 (12: hC13_CD8-MKI67) (Fig. 3F).

We next analyzed cell-cell communication networks that are enriched in the intrahepatic tumors compared to primary CRC tumors using CellChat analysis. These networks can be measured either in number or by strength, which is founded in the law of mass action[29]. All CellChat analysis was normalized for the frequency of cell types when creating the initial CellChat object. We analyzed aggregated cell-cell communication networks specifically to and from pTRT cells that were enriched in the intrahepatic TILs, represented by red connection vectors, as compared to primary CRC TILs, represented by blue connection vectors (Fig. 3G–J). Interactions between nonreactive CD8 + T cells to pTRT CD8 + T cells indicate an expected shared overlap between all CD8 + T cells. When analyzing broad cluster labeling, we were interested in identifying cell interactions highlighted in red (increased in liver metastasis compared to primary CRC) that were consistently sending and receiving signals from pTRT as characterized both by number and strength (Fig. 3G–J). We observed that pTRT showed a consistent pattern of sending and receiving signals to and from macrophages more in intrahepatic TILs than in primary CRC TILs in both frequency and strength (Fig. 3G–J). Furthermore, the same cell-cell interactions between macrophages and pTRT cells were more prominent in the metastatic tumor compared to the surrounding hepatic tissue (Supplementary Fig. 6A and B). We also found that the macrophage and pTRT cell-cell interaction was more pronounced in the surrounding hepatic tissue as compared to the surrounding colonic tissue (Supplementary Fig. 6C and D). Next, we studied macrophages in greater detail. When analyzing cluster labeling for macrophages at a granular level, pTRT cells showed increased communication to CXCL12 macrophages and CD3E monocyte clusters in intrahepatic TILS compared to primary CRC TILs (Fig. 3K). We also

observed that pTRT received increased communication from SPP1 as well as CD3E macrophage clusters (Fig. 3L). Next, we analyzed the cell-cell communication network from all macrophages to pTRT cells to identify pathways enriched in intrahepatic TILs as compared to primary CRC TILs (Fig. 3M). Across all macrophage clusters, SPP1 and FN1 ligands from macrophages interacted with CD44, as well as integrin receptors ITGA4_ITGB1 and ITGA4_ITGB7, on pTRT cells to a greater degree in the intrahepatic TILs as compared to primary CRC TILs (Fig. 3M).

In summary, a high frequency of pTRT CD8 + T cells was found in patients with hepatic metastasis. Additionally, intrahepatic pTRT cells preferentially interacted with SPP1 + and FN1 + macrophages. Previous studies have found SPP1 + macrophages to correlate with poor clinical outcomes in various cancer types[16,30,31]. SPP1 + expression in healthy liver has been reported to be minimal and mainly localized to Kupffer cells compared to other immune cells[32]. Fibronectin is a critical component of the extracellular matrix in liver fibrosis and has been linked to tumor recurrence after curative treatment[33,34]. Given these enriched interactions, the role of SPP1 + and FN1 + macrophages in TAS CD8 + T cell dysfunction was further studied.

### SPP1 + macrophages cause pTRT cell dysfunction

Previous human scRNA-seq analyses have shown that SPP1 + macrophages represent a larger proportion of macrophages in liver metastasis compared to primary colonic tumors[24]. We next profiled the macrophages in our murine models to see whether an enriched SPP1 and FN1-expressing macrophage population was also found in murine hepatic tumors. Mice were injected with CT26 tumor cells either intrahepatically or subcutaneously then intrahepatic and subcutaneous tumors were harvested after 15 days and TILs were analyzed by flow cytometry (Supplementary Fig. 7). There was a significantly higher frequency of Spp1 + macrophages in intrahepatic TILs compared to subcutaneous TILs (Fig. 4A). In addition, we found a significantly higher expression of PD-L1 on Spp1 + macrophages in the intrahepatic TILs compared to subcutaneous TILs (Fig. 4B). These Spp1 + macrophages were distinct from M2 phenotype (CD163 + CD206 +) which have protumor function and showed significantly decreased expression in the intrahepatic TILs as compared to subcutaneous TILs (Fig. 4C). Finally, the expression of Spp1 by macrophages showed a significant correlation with tumor weight (Fig. 4D). In summary, similar to the human scRNA-seq results, an enriched Spp1 macrophage population was seen in intrahepatic TILs which correlated with tumor size and showed a distinct phenotype from the M2 macrophages.

Previous studies have shown a correlation between hypoxia and SPP1 + macrophages[16]. Therefore, we utilized hypoxic conditions to induce a Spp1-positive phenotype in macrophages. Peritoneal macrophages were isolated (Supplementary Fig. 8A and B)[35] and placed under normoxic or hypoxic (0.5% $O_2$) conditions for 24 hours. Afterward, the cells were analyzed by flow cytometry. Under hypoxic

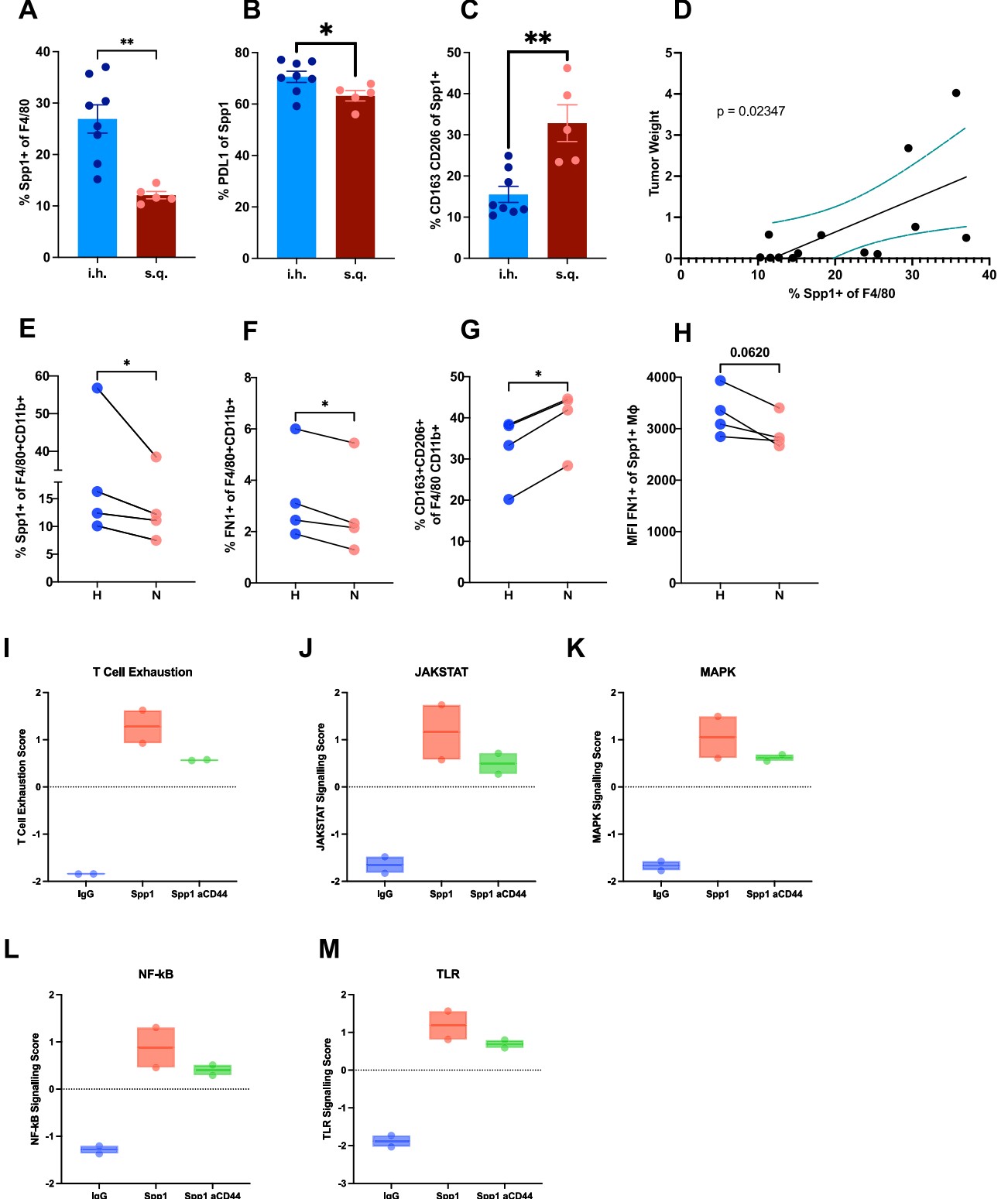

**Fig. 4 | Spp1 + Macrophages cause TAS CD8 + T cell dysfunction. A** Frequency of Spp1 + macrophages in intrahepatic ($n$ = 8 mice) and subcutaneous TILs ($n$ = 5 mice); $p$ = 0.0016. **B** PDL1 + frequency of Spp1 + macrophages in intrahepatic ($n$ = 8 mice) and subcutaneous TILs ($n$ = 5 mice); $p$ = 0.0428. **C** CD163 + CD206 + (M2) frequency of Spp1 + macrophages in intrahepatic ($n$ = 8 mice) and subcutaneous TILs ($n$ = 5 mice); $p$ = 0.0019. **D** Correlation between orthotopic tumors and frequency of Spp1 + macrophages ($n$ = 13). **E** Frequency of Spp1 + macrophages after hypoxia and normoxia ($n$ = 4 mice); $p$ =0.0186. **F** FN1 + frequency of macrophages after hypoxia and normoxia ($n$ = 4 mice); $p$ = 0.0466. **G** CD163 + CD206 + (M2) frequency of macrophages after hypoxia and normoxia ($n$ = 4 mice); $p$ = 0.0173.

**H** Spp1 + macrophage expression of FN1 (MFI) after hypoxia and normoxia ($n$ = 4 mice). Nanostring (**I**) T cell exhaustion, (**J**) JAKSTAT pathway, (**K**) MAPK, (**L**) NF-kB, and (**M**) TLR scores in IgG, Spp1, and Spp1 with anti-CD44 antibody ($n$ = 6; box plots showing both replicates as bounds of box and center indicating mean). Experiments were completed in female, BALB/c 6–8 week old mice (**A**–**H**) and female, C57BL/6 6–8 week old mice (**I**–**M**). Data was analyzed using ordinary one way Anova (**A**–**C**), two-sided paired $t$ test (**E**–**H**), simple linear regression (D): *$P$ < 0.05; **$P$ < 0.01; ***$P$ < 0.005; ****$P$ <0.001, ns = not significant (data are presented as mean values +/− SEM).

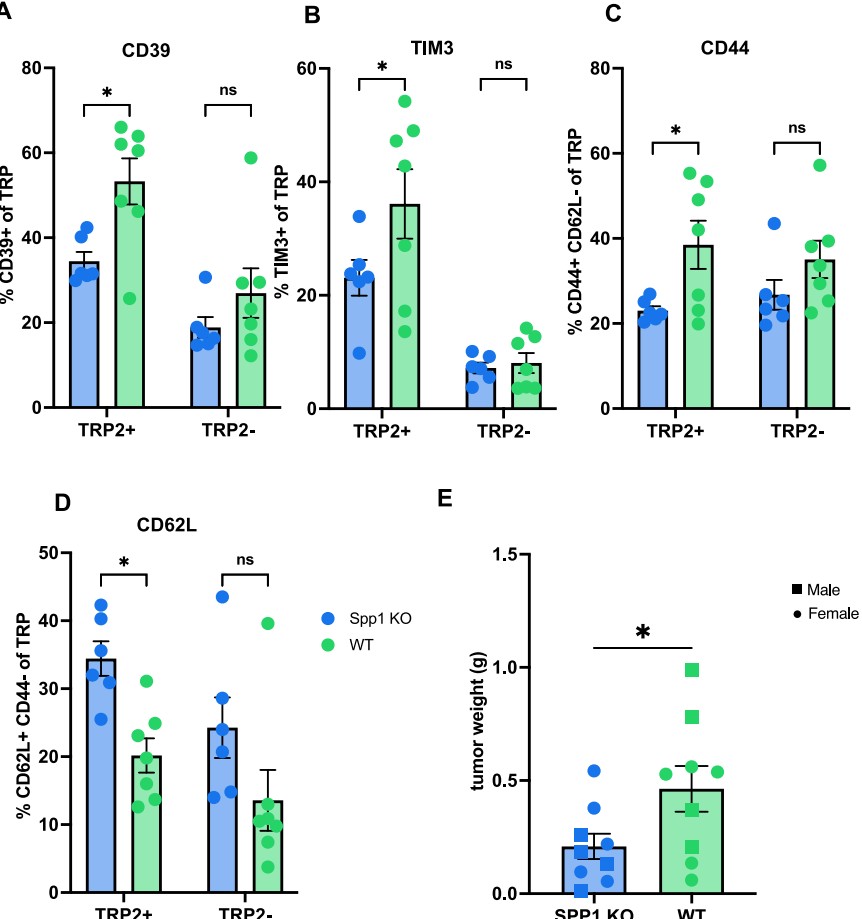

**Fig. 5 | Knock out Spp1 decreases TAS CD8 + T cell exhaustion and tumor burden.** Symbol legend for (**A**–**D**) can be found in (**D**). **A** Frequency of CD39 + TRP2 + (TAS) CD8 + T cells (left; $p = 0.0165$) and CD39 + TRP2- (TAS) CD8 + T cells (right; $p = 0.4001$) in Spp1 KO compared to WT mice. **B** Frequency of TIM-3+ TRP2 + CD8 + T cells (left; $p = 0.0464$) and TIM-3 + TRP2- CD8 + T cells (right; $p = 0.9834$) in Spp1 KO ($n = 6$) compared to WT mice ($n = 7$). **C** Frequency of CD44 + CD62L- TRP2 + CD8 + T cells (left; $p = 0.0338$) and CD44 + CD62L- TRP2- CD8 + T cells (right; $p = 0.3264$) in Spp1 KO ($n = 6$) compared to WT mice ($n = 7$). **D** Frequency of

CD44- CD62L + TRP2 + CD8 + T cells (left; $p = 0.0229$) and CD44- CD62L + TRP2- CD8 + T cells (right; $p = 0.0986$) in Spp1 knockout (KO) ($n = 6$) compared to WT mice ($n = 7$). **E** RIL-175 tumor weight comparing in Spp1 KO ($n = 5$) compared to WT mice ($n = 5$; $p = 0.0428$). Experiments (**A**–**E**) were completed in male and female B6.129S6(Cg)-Spp1tm1Blh/J 6–8 week old mice. Data was analyzed using 2-way Anova (**A**–**D**) and two-sided unpaired $t$ test (E): *$P < 0.05$; **$P < 0.01$; ***$P < 0.005$; ****$P < 0.001$, ns = not significant (data are presented as mean values +/− SEM).

conditions, macrophages expressed significantly higher levels of Spp1 (Spp1^high) compared to normoxic (Spp1^low) conditions (Fig. 4E and Supplementary Fig. 8C). In addition, under hypoxic conditions, macrophages had a significant increase in FN1 expression compared to normoxic conditions (Fig. 4F). Interestingly, similar to our findings in murine models, the expression of M2 markers (CD163 + CD206 +) on macrophages decreased under hypoxic conditions (Fig. 4G). Spp1 + macrophages expressed higher levels of FN1 under hypoxia, indicating co-expression of Spp1 and FN1 (Fig. 4H).

To investigate downstream signaling mechanisms specific to Spp1 contributing to T cell exhaustion, we incubated purified CD8 + T cells (Supplementary Fig. 8D) for 24 h under three conditions: (1) IgG control, (2) recombinant Spp1 protein, and (3) recombinant Spp1 protein with anti-CD44 antibody. Cells were analyzed using flow cytometry, and to further explore the molecular mechanisms underlying this exhaustion phenotype, we performed Nanostring nCounter® Immune Exhaustion Panel analysis. Probing into exhaustion under these isolated conditions, T cell exhaustion was increased upon Spp1 stimulation, as indicated by elevated CD39 expression and transcriptional T cell exhaustion score, and this effect was reversed by anti-CD44 treatment (Fig. 4I). Furthermore, downstream advanced analysis of the immune exhaustion panel included in the Nanostring panel

revealed key regulators of the exhaustion mediation. Genes associated with JAK-STAT, MAPK, NF-κB, and TLR signaling pathways were upregulated in response to Spp1 stimulation. Notably, the addition of soluble anti-CD44 reversed these changes, suggesting that Spp1-CD44 interactions activate multiple exhaustion-related pathways in CD8 + T cells (Fig. 4J–M). These findings support the role of Spp1 in modulating T cell exhaustion at a transcriptional level.

To determine the effect of Spp1 + macrophages in vivo, we implanted intrahepatic B16F10 tumors in both homozygous Spp1 knockout (KO) and wild-type (WT) mice. Flow cytometric analysis was performed on day 15. We found that TAS CD8+ T cells from Spp1 KO mice had decreased expression of exhaustion markers, CD39 and TIM-3, as compared to WT mice (Fig. 5A and B). Moreover, this difference was only observed in TAS (TRP2+) CD8 + T cells and not in TRP2- CD8 + T cells (Fig. 5A and B). CellChat analyses of human T cells and macrophages showed an interaction of SPP1 on macrophages with CD44 on pTRT CD8 + T cells (Fig. 3M), leading us to investigate whether a similar observation could be made in mice. Compared to WT mice, CD44 + CD62L- expression was also decreased on TAS CD8 + T cells, while the naïve (CD44- CD62L +) phenotype was increased in Spp1 KO mice (Fig. 5C, 5D). Similar to the exhaustion markers, these changes were significant only for TRP2+ CD8+ T cells rather than TRP2- C8 +

T cells (Fig. 5C, D). Finally, to examine the effect of SPP1 on overall tumor burden, we injected a mouse HCC cell lines (RIL-175) orthotopically into the livers of Spp1 KO and WT mice. Mice were sacrificed after 20 days (RIL-175) post-injection, and tumor weights were measured. Knocking down Spp1 significantly decreased tumor sizes with RIL-175 (Fig. 5E).

In summary, our findings show that SPP1-CD44 interactions were enriched in the liver of stage IVA CRC patients based on scRNA-seq analysis. Subsequently, we found that Spp1+ macrophages promote TAS CD8 + T cell exhaustion and dysfunction, affecting tumor burden. Additionally, we showed Spp1 + macrophages to co-express FN1 (Fig. 4F, H). To better understand the effect of fibronectin co-expression, we next explored the role of extracellular matrix (ECM) changes in the liver TME.

### Macrophage and CD8 + T cell interactions are enriched in αSMA + environments

Previous studies have shown SPP1+ macrophages interact with fibroblasts[36,37]. However, based on the aforementioned CellChat analysis, we postulate macrophage induced differences in ECM components, including fibronectin, affect macrophage-CD8 + T cell interactions. We decided to study the spatial interaction of macrophage-CD8 + T cell interaction in liver tumors. We recently performed a spatial co-detection by indexing (CODEX) imaging analysis as well as scRNA-seq analysis from liver tumors and adjacent tissue of HCC patients[38]. Before using this data set here, we repeated scGSEA on this HCC scRNA-seq data set with the same approach as our previous scRNA-seq analysis to define pTRT CD8 + T cells in these HCC patients (Fig. 3A). CellChat analysis of our RNA-seq data confirmed that SPP1 + macrophage interactions with pTRT cells also occurred in HCC (Supplementary Fig. 9A)[38]. The CODEX imaging identified cell types at a single-cell resolution using our previously published thresholds to define the positive population for each marker (Fig. 6A)[38]. This high-dimensional, integrated multi-omic analysis of protein (CODEX) and mRNA expression (scRNA-seq) connects intercellular signaling on a spatial and transcript level in the context of a single pathology.

FN1 expression in the ECM mediates TGF-β induced fibroblast contraction and αSMA expression[39]. Thus, we annotated our CODEX images for regions that were both rich in αSMA and morphologically overlapped with a profibrotic structure (Fig. 6B, C). We then analyzed at a single cell resolution, CD8 + T cells and macrophages in profibrotic (αSMA +) regions compared to the surrounding tissue (αSMA-neg) regions (Fig. 6D, E). Cell quantification analysis showed no difference between macrophage density between αSMA+ and αSMA-neg regions (Fig. 6F). However, proximity analysis revealed that macrophages were closer to CD8 + T cells in αSMA + compared to αSMA-neg regions (Fig. 6G). In addition, macrophages were closer to proliferating (Ki67 +), CD45RA +, CD45RO +, and exhausted (CD39 +) CD8 + T cells in the αSMA+ regions compared to αSMA-neg regions (Fig. 6H-K). As a negative control for the next proximity analysis, there was no difference between the proximity of macrophages to NK cells between αSMA + and αSMA- regions (Fig. 6L). In addition, no differences were observed between CD8 + T cells and NK cells within αSMA + and αSMA-neg regions (Fig. 6M). Similar results were found for Tregs (Fig. 6N and O).

In summary, we investigated the role of profibrotic ECM changes in the liver TME by analyzing CODEX data of HCC patients. We found that CD8+ T cells, including proliferating, CD45RA +, CD45RO + and exhausted phenotypes, were more likely to be in close proximity to macrophages in profibrotic αSMA + regions than in αSMA-neg regions. We next aimed to reconcile the profibrotic interactions with the earlier flow cytometry results from the livers of mice with subcutaneous tumors.

### Profibrotic environment enriches SPP1

Thus far, both CODEX and CellChat results showed that macrophage-CD8 + T cell interactions were enriched in profibrotic, αSMA + regions.

In addition, time course experiments showed on day 21 an increase in CD39 (Figs. 2A and 7A), TIM-3 (Fig. 7B and Supplementary Fig. 3A), PD1 (Fig. 6C and Supplementary Fig. 3G), and CD69 (Fig. 6D and Supplementary Fig. 3D) expressing TAS CD8 + T cells in the liver of mice with subcutaneous tumors. We next asked whether increased profibrotic machinery in the liver could explain why there was such a high frequency of exhausted, activated TAS CD8 + T cells in the liver of mice with subcutaneous tumors.

TGF-β and IL-6 serve as important regulators of inflammation and fibrosis, and their production and action are potentiated by upstream TLR-2 and TLR-4[40–45]. These are expressed on a wide variety of cells, including hepatocytes, hepatic stellate cells, biliary epithelial cells, liver sinusoidal endothelial cells, dendritic cells, and Kupffer cells[40,41].

We questioned if the extrahepatic tumor could be upregulating profibrotic machinery in the liver. In addition, we were interested in studying if the primary CRC tumor controls the liver TME in the same manner that it affects the local colonic TME. To answer this, we went back to the human patient scRNA-seq data to correlate the intrahepatic SPP1 and FN1 response to profibrotic proteins in primary CRC tumors.

Interestingly, a significant correlation was found between the expression of TGF-β and IL-6 expression in primary CRC tumor and FN1 in the metastatic intrahepatic tumor (Fig. 7E) but not the local primary CRC tumor (Fig. 7F). Similar data was also observed for TLR-2 and TLR-4 (Supplementary Fig. 9B–E). Thus, extrahepatic tumor production of profibrotic proteins correlated with intrahepatic expression of fibronectin but not with local fibronectin.

We next asked whether there was a correlation between SPP1 expression by macrophages in the profibrotic environments and TGF-β, IL-6, TLR-2, and TLR-4 expression levels in the primary CRC tumors of the patients. Interestingly, there was a significant correlation between TGF-β, IL-6, TLR-2, and TLR-4 in the primary CRC tumor and SPP1 expression by macrophages in the metastatic intrahepatic tumor (Fig. 7G and Supplementary Figs. 9F, G). However, no such correlation to the local primary CRC tumor was found (Fig. 7H and Supplementary Fig. 9H, I). In addition, no significant correlation was observed between M2 macrophage markers CD163 and CD206 in the intrahepatic tumor to TGF-β, IL-6, TLR-2, and TLR-4 in the primary CRC tumor (Supplementary Fig. 9J–Q). In summary, extrahepatic tumor production of profibrotic proteins correlated with intrahepatic SPP1 expression but no correlation to intrahepatic M2 markers or local SPP1 expression.

In summary, we demonstrate a connection between TGF-β, IL-6, TLR-2, and TLR-4 expression by the primary CRC tumor to SPP1 and FN1 expression in the metastatic intrahepatic tumor. Expression of these cytokines did not correlate with the surrounding colonic microenvironment. We next analyzed the macrophage population to further elucidate the mechanism by which these cytokines preferentially affect the liver.

### Pseudotemporal Analysis Reveals a Liver-Specific Intermediate Cell Population

Using the previous annotated scRNA-seq data of paired CRC liver metastasis, 4833 macrophage and monocyte cells in the liver were clustered (Fig. 8A)[24]. Kupffer cell (KC) markers, VSIG4 and CSF1R, were found to be highly co-expressed in the top region of the UMAP (Supplementary Fig. 10A). We utilized pseudotime trajectory analysis of the intrahepatic macrophage and monocytes cells to infer the relationship of SPP1, TGF-βR, and IL-6R to dynamic, continuous gene changes (Fig. 8B).

Of all 24,622 genes tested, SPP1 was the 4th highest correlation to a function of pseudotime (Moran's I = 0.72) (Supplementary Fig. 10C). SPP1 trajectory analysis revealed differentiation from the branch point along the upper leaf of pseudotime (Fig. 8C). Highest SPP1 expression was found at the terminal end of pseudotime trajectory analysis (Fig. 8C). This leaf of pseudotime was analyzed for continual

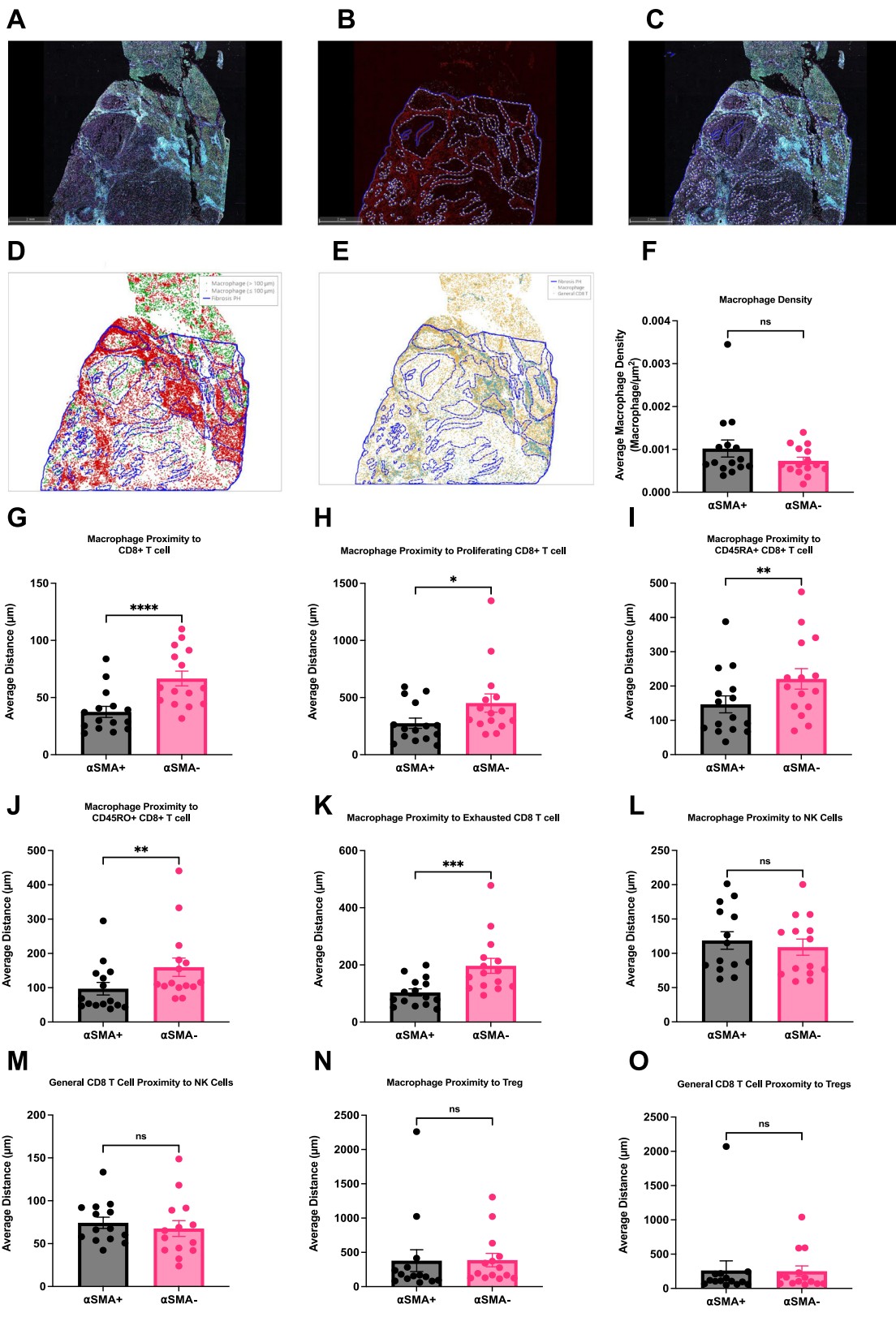

changes in the expression of relevant markers earlier in the trajectory (Fig. 8D–F). TGF-βR1 and IL-6 were found to closely follow the trajectory and were expressed earlier in pseudotime as compared to SPP1 (Fig. 8D). Similar trajectory results were found for KC markers CSF1R and VSIG4 (Fig. 8E) as well as CD163 and CD206 (Fig. 8F). Combining pathway analysis, scGSEA, and trajectory analysis, genes that upregulate collagen production in the surrounding microenvironment were analyzed (Supplementary Fig. 10D). Collagen upregulation genes were expressed at an earlier pseudotime in the trajectory analysis as compared to SPP1. Throughout this analysis, we detected a portion of the cells on the UMAPs that co-expressed all these markers at an intermediate pseudotime. Thus, we next clustered all cells along this leaf of pseudotime and found 3 distinct populations (Fig. 8G).

**Fig. 6 | CODEX analysis of macrophage and CD8 + T cell interactions in αSMA rich regions. A** Overview of whole-tissue section of human HCC sample with 37-plex CODEX panel. **B** Annotations (blue) of profibrotic αSMA + (red) regions. **C** Overlay of annotations (**B**) on whole-tissue section (**A**). **D** Single cell resolution annotation of macrophages and CD8 + T cells. **E** Representative spatial plot of macrophages proximity analysis to CD8 + T cells where macrophages within 100 μM of CD8 + T cells are labeled in red and macrophages farther than 100 μM of CD8 + T cells are labeled in green. **F**) Macrophage density in αSMA + and αSMA-regions (*n* = 15 patients; *p* = 0.0957). **G** Macrophage proximity to CD8 + T cells in αSMA + and αSMA- regions (*n* = 15 patients; *p* < 0.0001). **H** Macrophage proximity to Ki67 + CD8 + T cells in αSMA + and αSMA- regions (*n* = 15 patients; *p* = 0.0127).

**I** Macrophage proximity to CD45RA + CD8 + T cells in αSMA + and αSMA- regions (*n* = 15 patients; *p* = 0.0014). **J** Macrophage proximity to CD45RO + CD8 + T cells in αSMA + and αSMA- regions (*n* = 15 patients; *p* = 0.0032). **K** Macrophage proximity to CD39 + CD8 + T cells in αSMA + and αSMA- regions (*n* = 15 patients; *p* = 0.0004). **L** Macrophage proximity to NK cells in αSMA + and αSMA- regions (*n* = 15 patients; *p* = 0.5322). **M** CD8 + T cell proximity to NK cells in αSMA + and αSMA- regions (*n* = 15 patients; *p* = 0.4420). **N** Macrophage proximity to Treg in αSMA + and αSMA- regions (*n* = 14 patients; *p* = 0.9391). **O** CD8 + T cell proximity to Tregs in αSMA + and αSMA- regions (*n* = 14 patients; *p* = 0.9003). Data was analyzed by two-sided paired *t* test (F-O): *$p < 0.05$; **$P < 0.01$; ***$P < 0.005$; ****$P < 0.001$, ns = not significant (data are presented as mean values +/− SEM).

A KC population (Fig. 8G) was found that highly expressed CD163 (Fig. 8H), CD206 (Fig. 8I), VSIG4 (Fig. 8J), CSF1R (Fig. 8K), and IL-18 (Fig. 8O). This KC population also highly expressed previously published gene scores for identifying KCs (Fig. 8R)[46]. The KC population had lower expression of SPP1 (Fig. 8L), IL6R (Fig. 8M), TGF-βR1 (Fig. 8N), positive collagen regulation (Fig. 8Q). These cells were found at an earlier pseudotime (Fig. 8P).

A cluster of macrophages (Fig. 8G) highly expressed SPP1 (Fig. 8L). This SPP1 + macrophage population downregulated CD163 (Fig. 8H), CD206 (Fig. 8I), VSIG4 (Fig. 8J), CSF1R (Fig. 8K), IL-18 (Fig. 8O), and KC score (Fig. 8R). This SPP1 macrophage population, which distinctly clustered away from the KC population, was found at the end of pseudotime (Fig. 8P).

Recent literature identifies SPP1 macrophages as distinct from KCs in fatty liver[47]. In contrast, in the liver TME, an intermediate cluster (IC) population (Fig. 8G) was found to co-express intermediate levels of CD163 (Fig. 8H), CD206 (Fig. 8I), VSIG4 (Fig. 8J), CSF1R (Fig. 8K), SPP1 (Fig. 8L), IL6R (Fig. 8M), TGF-βR1 (Fig. 8N), IL18 (Fig. 8O), and KC score (Fig. 8R). This cluster had the highest expression of IL6R (Fig. 8M) and TGF-βR1 (Fig. 8N) as well as positive collagen regulation (Fig. 8Q). This cluster was isolated in an intermediate pseudotime state bridging the SPP1 and KC clusters (Fig. 8P).

Performing a similar analysis for colonic macrophages and monocytes in paired CRC metastasis, 1330 macrophage and monocyte cells were clustered (Supplementary Fig. 10E)[24]. Resident macrophage markers VSIG4 and CSF1R were found to be highly co-expressed in the left region of the UMAP (Supplementary Fig. 10B). We utilized pseudotime trajectory analysis of the intrahepatic macrophage and monocytes cells to infer the relationship of SPP1, TGF-βR, and IL-6R to dynamic, continuous gene changes (Supplementary Fig. 10F). SPP1 was not correlated with pseudotime in the colon (Moran's I = 0.29) (Supplementary Fig. 10G). Unlike the liver, no significant relationship was found between TGF-βR1, IL6R, CSF1R, VSIG4, CD163, CD206, and pseudotime (Supplementary Figs. 10H–K). When comparing the frequency of cells co-expressing SPP1, TGF-βR1, IL6R, CSF1R, VSIG4, and CD163 per patient, the liver had a significantly larger IC population (Fig. 8S). Bulk TCGA analysis with ssGSEA found that patients with enriched IC expression had worse survival in HCC but not in colon cancer (Supplementary Fig. 10L). In summary, an expanded IC population unique to the liver expressing high levels of TGF-βR1 and IL-6R was found later in pseudotime compared to KCs.

To validate the above findings of a premetastatic niche in our murine models and liver metastasis, we next analyzed a third, recently published scRNA-seq data of patients with pancreatic adenocarcinoma[48]. Biopsies of the non-tumor-bearing liver at the time of resection were analyzed using scRNA-seq (Fig. 10M), and these patients were then followed to determine the incidence of liver metastasis. After QC, 26,627 total cells were plotted on a UMAP (Supplementary Fig. 10M). 2 clusters containing 2345 cells were found to express CD68 (Supplementary Fig. 10M). These clusters were then analyzed for IC scGSEA enrichment (Fig. 8T). IC enrichment was detected in patients who would develop early and late liver metastasis

(Fig. 8T) as compared to control (non-pancreatic adenocarcinoma) or extrahepatic metastasis patients. Individually, VSIG4, CSF1R, and SPP1 were not highly predictive in patients who will develop liver metastasis in the future (Supplementary Fig. 10N). In summary, the IC population was enriched in the premetastatic niche, which may predict future liver metastasis.

Given that the response to profibrotic factors is mainly observed in the liver rather than the primary CRC tumor, we next probed the fibronectin levels in the metastatic intrahepatic tumor in the human scRNA-seq data. The metastatic intrahepatic tumor had higher FN1 expression (Fig. 8U). Thus far, we have shown a correlation between profibrotic proteins in the primary CRC tumor and the expression of FN1 and SPP1 in the metastatic intrahepatic tumor.

We next questioned whether mice with subcutaneous tumors also had changes in liver ECM components. Fibronectin is necessary for collagen matrix assembly, and while many types of collagen make up liver fibrosis ECM, collagen type 4 has been a valuable biomarker[34,49]. Collagen type 4 levels were studied utilizing qPCR to investigate the effect of these profibrotic changes on the polarization of ECM components. This was chosen over H & E staining as no morphological changes in the liver architecture were expected. Subcutaneous tumor-bearing mice as well as liver tumor-bearing mice (tumors were injected into the right liver lobe) were compared to tumor-free mice on day 15. On day 15, the left portion of the liver contralateral to the tumor site was harvested, and type 4 collagen levels were analyzed using RT-qPCR. Liver tumor-bearing mice showed a 10-fold increase in type 4 collagen in the non-tumor liver (Fig. 8V). Additionally, when tumors were implanted in the subcutaneous space, an increase in collagen type 4 was found in the liver (Fig. 8V).

In order to probe the metastatic tumor burden in a different metastasis model in mice, we injected the RIL175 cells into the spleens of Spp1 KO and WT mice to study a liver metastasis model[50]. Mice were sacrificed after 18 days post-injection, and liver metastatic burden (presence and number of tumor nodules) was measured. Spp1 KO had significantly decreased metastatic liver tumor burden (Fig. 9A and B).

In summary, both local and distal tumor signaling correlated to the profibrotic polarization and enrichment of an IC population in the liver, as found through scRNA-seq of patient samples and qPCR of mouse livers.

## Discussion

We studied tumor-reactive CD8 + T cells using a high-dimensional, integrated approach utilizing three human scRNA-seq data sets (including appropriate per-patient analysis), human CODEX imaging, murine models, in vitro assays, and ex vivo systems in livers and primary tumors[51]. Both in mice and patients, we found an unexpectedly high frequency of tumor-reactive CD8 + T cells in liver metastases. Consistent with previous studies, we found that tumor-reactive CD8 + T cells displayed a dysfunctional state within the liver TME in both murine models and human scRNA-seq data[20,21]. Our comprehensive analysis of the human scRNA-seq data indicated a variable phenotype of tumor-reactive CD8 + T cells across different organs. Furthermore,

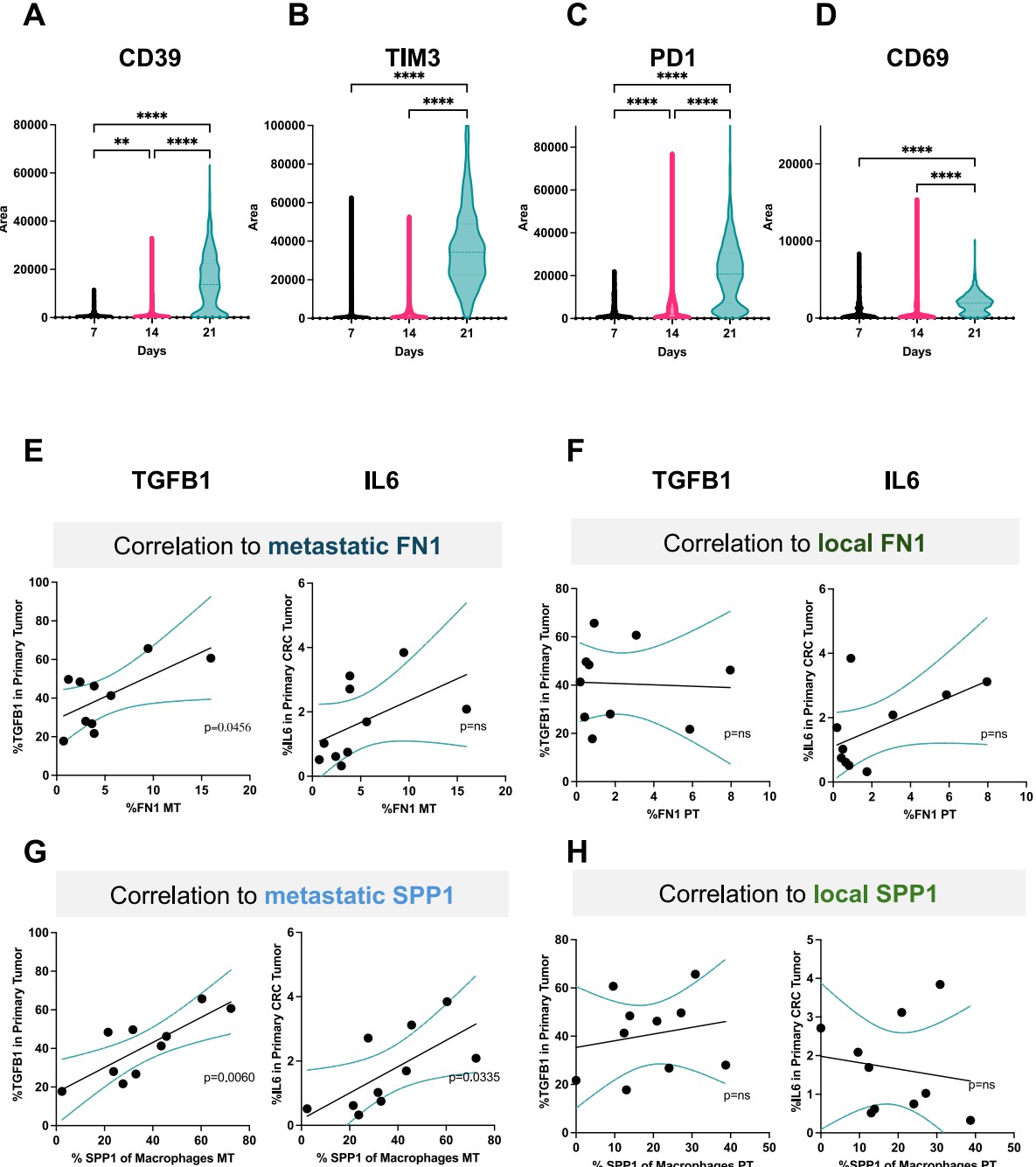

**Fig. 7 | Exhausted, profibrotic polarization of the liver correlated to extrahepatic tumor signaling.** AH1 + CD8 + T cells in the liver of subcutaneous tumor-bearing mice expressing CD39 (**A**; day 7 vs 14 $p = 0.0083$; day 7 vs 21 $p < 0.0001$; day 14 vs 21 $p < 0.0001$), TIM3 (**B**; day 7 vs 14 $p = 0.0760$; day 7 vs 21 $p < 0.0001$; day 14 vs 21 $p < 0.0001$), PD1 (**C**; day 7 vs 14 $p < 0.0001$; day 7 vs 21 $p < 0.0001$; day 14 vs 21 $p < 0.0001$), and CD69 (**D**; day 7 vs 21 $p < 0.0001$; day 14 vs 21 $p < 0.0001$). **E** Correlation of fibronectin expression in metastatic intrahepatic tumors (MT) to TGF-β and IL-6 in primary tumors ($n = 10$). **F** Correlation of fibronectin expression in primary CRC tumors (PT) to TGF-β and IL-6 in primary tumors ($n = 10$). **G** Correlation of SPP1 expression by macrophages in metastatic intrahepatic tumors (MT) to TGF-β and IL-6 expression in primary tumors ($n = 10$). **H** Correlation of SPP1 expression by macrophages in primary CRC tumors to TGF-β and IL-6 in primary tumors ($n = 10$). Data was analyzed with simple linear regression (**E–H**) and one-way Anova (**A–D**): *$P < 0.05$; **$P < 0.01$; ***$P < 0.005$; ****$P < 0.001$, ns = not significant.

SPP1 + macrophages were found to interact with tumor-reactive CD8 + T cells in humans. These SPP1 + macrophages are distinct from the M2 macrophages in vivo and in vitro, consistent with previous studies[16]. Additionally, tumor signaling was seen to affect profibrotic polarization in the liver, enriching the SPP1 interactions and contributing to CD8 + T cell dysfunction in the tumor microenvironment. This signaling enriched an IC population that was found to connect KC and SPP1 macrophage populations using pseudotime trajectory analysis

and positively regulate collagen production. Furthermore, analysis of the healthy liver of subcutaneous tumor-bearing mice revealed elevated collagen production.

The presence of SPP1 + macrophages correlates with worse survival in various cancer types[16,30,31]. However, our understanding of their biological function in the TME and their effects on tumor-specific T cell responses remains limited[16,37,47]. For decades, M2 macrophages have been recognized as cells that promote tumor growth[52,53]. Both

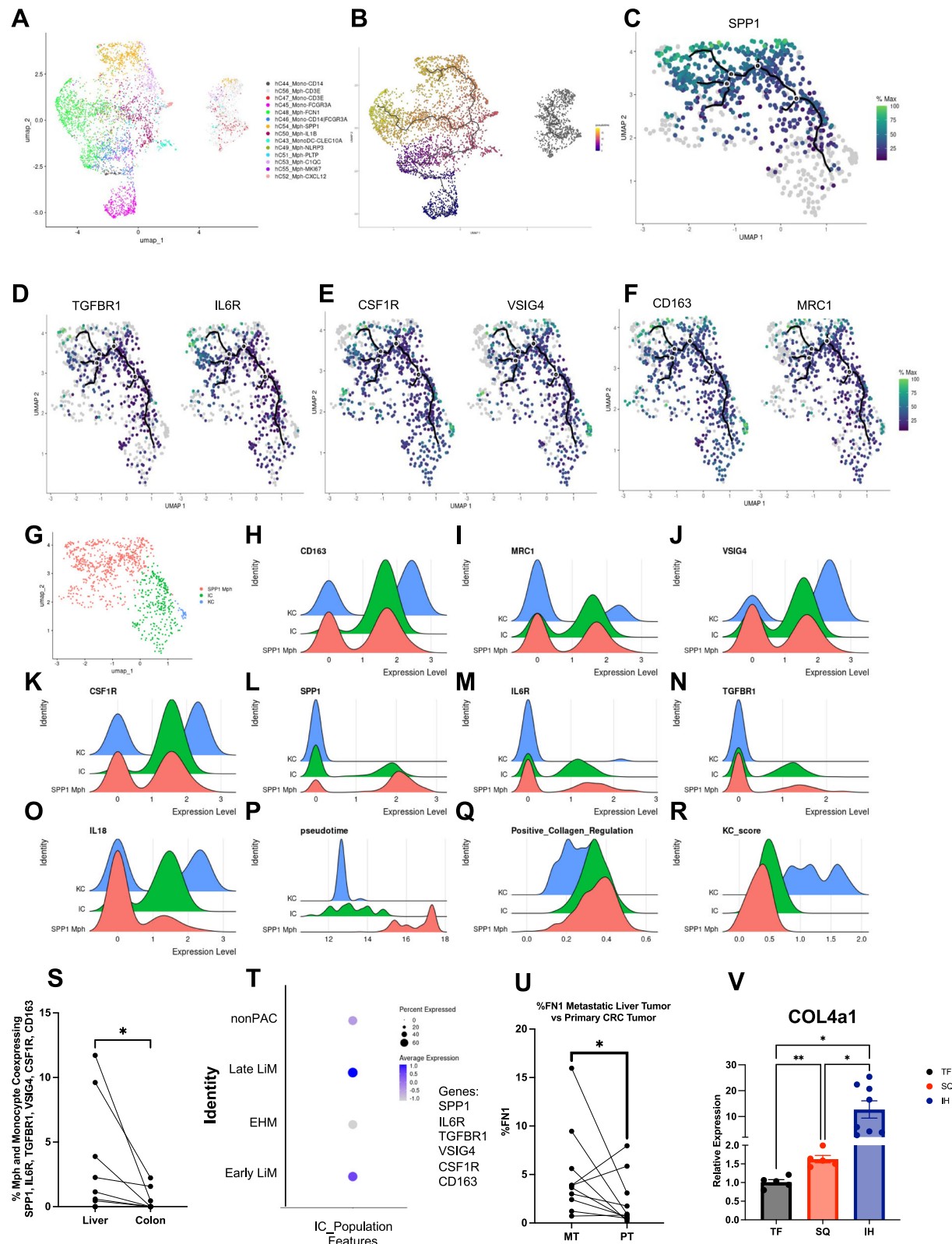

functional murine in vivo studies and comprehensive bioinformatic analysis of human scRNA-seq data on T cells and M2 macrophages have provided findings complementary to the protumor role of SPP1 + macrophages[8,15]. Previous studies have shown the potential role of SPP1 + macrophages on effector T cells, hypothetically separate from the M1/M2 dogma[16]; however, there is limited understanding of whether SPP1 + macrophages can induce tumor-reactive CD8 + T cell

exhaustion and dysfunction without an enriched M2 macrophage population[16,36,37,54,55]. Previous studies on both T cell graveyard and SPP1 interactions have focused on intrahepatic mediators. The T cell graveyard or responder trap phenomenon describes an accumulation of apoptotic, activated CD8 + T cells in the liver[56,57]. This mechanism is mediated by Kupfer cell (KC) expression of FasL & nitric oxide, CD8 + T cell upregulation of LFA-1, antigen presentation by liver sinusoidal

**Fig. 8 | Intermediate cell population secondary to metastatic signaling.** Intrahepatic macrophage and monocyte population UMAP from metastatic CRC tumor-bearing liver, including macrophages (Mph) and monocytes (Mono) (**A**) (*n* = 10). Pseudotime trajectory analysis (*n* = 10) of intrahepatic macrophages and monocytes (**B**). Top branch of pseudotime analysis showing correlation of SPP1 (**C**), TGF-βR1 and IL6R (**D**), CSF1R and VSIG4 (**E**), and CD163 and CD206 (**F**) with pseudotime trajectories (*n* = 10). Subclustering of pseudotime branch related to SPP1 (*n* = 10) (**G**) with ridge plot displaying expression of CD163 (**H**), CD206 (**I**), VSIG4 (**J**), CSF1R (**K**), SPP1 (**L**), IL6R (**M**), TGF-βR1 (**N**), IL18 (**O**), pseudotime (**P**), positive collagen regulation score (**Q**), Kupfer cell (KC) score (**R**). **S** Frequency of cells coexpressing all markers of IC cluster in all macrophages and monocytes in the liver and the

colon per patient (*n* = 10; *p* = 0.0156). **T** Intermediate cluster (IC) score calculated in CD68 + per group in non-pancreatic adenocarcinoma cancer patients (nonPAC), late liver metastasis (Late LiM), extrahepatic metastasis (EHM), early liver metastasis (early LiM). **U** Expression of fibronectin in metastatic intrahepatic tumor compared to primary CRC tumor (*n* = 10 patients; *p* = 0.0268). **V** Collagen type IV subunit A qPCR expression in the liver of tumor-free (*n*=5), subcutaneous tumor (*n*=5), and intrahepatic tumor-bearing (*n* = 8) 6–8 week old female, BALB/c mice (TF vs SQ *p* = 0.0037; TF vs IH *p* = 0.0276; SQ vs IH *p* = 0.0353). Data were analyzed with two-sided paired *t* test (S,U) and one-way anova (V): *P< 0.05; **P< 0.01; ***P< 0.005; ****P< 0.001, ns = not significant (data are presented as mean values +/− SEM).

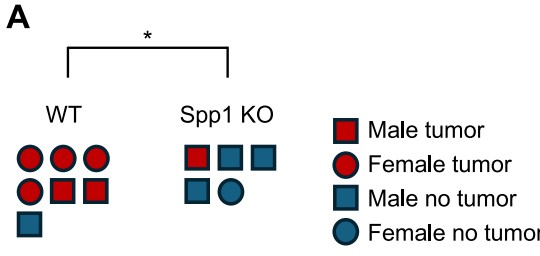

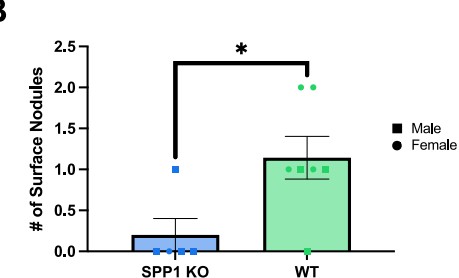

**Fig. 9 | Spp1 knock out decreases metastatic burden. A** Presence of RIL-175 metastatic liver tumor comparing in Spp1 KO (*n* = 5) compared to WT mice (*n* = 7); *p* = 0.0289. **B** Number of metastatic surface nodules comparing Spp1 KO (*n* = 5) compared to WT mice (*n* = 7); *p* = 0.0238. Experiments (**A**, **B**) were completed in

male and female B6.129S6(Cg)-Spp1tm1Blh/J 6–8 week old mice. Data was analyzed using two-sided unpaired *t* test (A-B): *P< 0.05; **P< 0.01; ***P< 0.005; ****P< 0.001, ns = not significant (data are presented as mean values +/− SEM).

endothelial cells (LSEC) & hepatocyte, and intrahepatic expression of IL-2, IL-4, IL-12, IL-15, and IL-18[56–58]. In contrast, we describe a tumor- and liver-specific mechanism where extrahepatic tumor signals cause the overexpression of an intermediate macrophage cell population. This intermediate macrophage population then upregulates collagen deposition and enriches immunosuppressive SPP1 + macrophage-CD8 + T cell interactions.

SPP1 macrophages have been found to interact with fibroblasts in the colon and CRC metastasis in the liver[36,37]. SPP1 ligand and PD-L1 on SPP1 macrophages promote CD8 + T cell exhaustion in vitro as well as in lung adenocarcinoma and gastric cancer[54,55,59]. Recent studies have found SPP1 macrophages to be a distinct macrophage subset from KCs in fatty liver[47]. Rather than a SPP1 population expanding only after seeding of metastatic cancer cells, we postulate a mechanism linking primary extrahepatic tumors to premetastatic expansion of an intrahepatic TGF-βR+ IL6R+ SPP1-KC intermediate macrophage population.

Previous literature found in fatty liver that these SPP1 macrophages were distinct from KC and were derived from monocytes[47]. This study provides insight into their role within the liver TME including a unique mechanism of their development. Here, we find that SPP1 macrophages were KC-derived through TGF-β and IL6 signaling. In addition, IC enrichment in the premetastatic niche may predict liver metastasis by promoting tumor-specific CD8 + T cell exhaustion and enrichment of fibrosis in the TME.

Multi-omic scRNA-seq-CODEX analysis of livers indicated that SPP1 + macrophages and CD8 + T cells interact in a profibrotic liver environment. Time course experiments revealed CD8 + T cell exhaustion in the liver of mice with subcutaneous tumors prior to tumor metastasis. This finding led us to hypothesize that primary tumors may promote the formation of profibrotic areas in the liver and explain the preferential formation of metastasis in the liver. Previous studies have found key chemokines and ligands involved in both metastasis of other cancer types as well as liver fibrosis[60–62].

Specifically, TGF-β and IL-6 are one of the few cytokines overlapping in both fields of study[60–62]. TGF-β has previously been found in both prostate cancer and breast cancer to induce distant bone metastasis by establishing driver gene mutations that create a premetastatic niche in distant tissue[60,61]. Similarly, IL-6 was identified as a mediator for crosstalk between bone marrow and cancer cells in breast cancer models. Levels of IL-6 correlated with increased monocyte dendritic progenitor growth along with an increased incidence of metastasis[63]. TLR-4 potentiates TGF-β production and action, while TLR-2 functions similarly to IL-6[42–45]. Our data suggests that these profibrotic proteins are signaling to the distant liver, enriching an intermediate cluster correlating to fibronectin and SPP1 expression and inducing CD8 + T cell exhaustion and dysfunction.

The size of the of human samples used in our scRNA-seq analysis was relatively modest, and stratification based on survival for these samples was not possible as such data is not available at this time. However, three distinct scRNA-seq data sets were used in this analysis to further address these issues as best possible. In addition, further research is needed on the effect of immune checkpoint blockade on the SPP1 + macrophage-CD8 + T cell interactions. Finally, this study had limited mechanistic data on the relationship between the profibrotic proteins and liver fibrosis since these specific mechanisms are discussed elsewhere in the literature, including the large number of downstream secreted chemokines after receptor engagement[45,64].

Future work is needed to study other organ sites enriched for SPP1 + macrophages-tumor-reactive CD8 T cells interactions. The SPP1 + macrophage-CD8 T cell interaction is a bidirectional, targetable interaction, addressing an urgent need for a new avenue for potential drug development. Separately, we found changes to the fibrotic polarization in the liver TME causing tumor-reactive CD8 T cell dysfunction in advance of tumor spread. This result provides a secondary thread of interest for further research on the obvious clinical implications of reducing metastatic burden to the liver through reversal of this profibrotic machinery. In summary, here, we describe a

mechanism by which primary tumors control distant CD8 T cells function by supporting the interaction of CD8 + T cells with SPP1 + macrophages in pro-fibrotic areas of the liver.

## Methods

### Mice

All technical procedures and experimental endpoints regarding mouse experiments were approved by the NCI Division of Intramural Research Animal Care and Use Committee. Mice were housed at the CRC animal facility at NCI in accordance with the National Research Council Guide for the Care and Use of Laboratory Animals. All animal protocols followed were developed in accordance to the PHS Policy on Humane Care and Use of Laboratory Animals. These were submitted and approved by the NIH prior to being performed. Mice were kept in isolation cages, at a maximum of 5 mice per cage. These were placed on stable temperature, humidity, and light cycle periods of 12 h in accordance with their circadian rhythm. Four-to-eight-week-old male and female C57BL/6 mice (strain code #000664) were purchased from The Jackson Laboratory. Female BALB/c mice (strain code #028) were purchased from Charles River Laboratories. Four-to-eight-week-old female and male B6.129S6(Cg)-Spp1tm1Blh/J (OPN KO) (strain code #004936; RRID:IMSR_JAX:004936) and B6(Cg)-Tyrc-2J/J (strain code #000058) were purchased from The Jackson Laboratory.

### Tumor models

The following murine tumor cell lines were used: B16-F10[65], CT26 (ATCC, Cat #CRL-2638, RRID:CVCL_7256), and RIL-175[66]. All cell lines tested negative for Mycoplasma.

Regarding humane endpoints for experiments, animals with necrotic tumors, had cutaneous ulceration, multiple tumors that weighed in total more than 10% of the animal's body weight, or presented with abdominal distension were euthanized, thereby ending the experiment. These criteria, including maximal tumor size/burden, were never exceeded.

For i.h. tumor establishment, cell suspensions of $2 \times 10^5$ tumor cells were prepared and resuspended in a 1:1 mix of PBS and Matrigel (Corning, Cat. No. 354230). Intrahepatic tumors were established by injecting cells in a total volume of 20 μL into the left lateral liver lobe[67]. Intrasplenic tumor injection model: To induce metastatic liver tumors, intrasplenic injections of RIL-175 tumor cells were performed as we previously described[50,68,69]. Briefly, the mouse spleen was exposed under anesthesia, and $5 \times 10^5$ B16-F10 cells in 250 μL layered solution of PBS were injected into the spleen. 18 days later, mice were euthanized, and liver tumors were counted. Subcutaneous models: $2 \times 10^5$ cells were injected subcutaneously in a total volume of 100 μl into the right flank. Tumor sizes were measured twice per week and calculated as $S = length \times width$. $2 \times 10^5$ cells were injected iv by the tail vein to induce lung metastasis.

**Cell Isolation.** Isolation of liver mononuclear cells and tumor-infiltrating lymphocytes (TILs) from tumor-bearing liver has been described previously[67]. Briefly, livers were removed immediately after the mice were sacrificed. Tumors from tumor-bearing livers were separated from the liver to be processed independently. After homogenization, debris was removed by passing through nylon mesh. Liver infiltrating cells were isolated by isotonic Percoll centrifugation (850xg, 25 min). Red blood cells were lysed by using ACK lysing buffer. For macrophage analysis of liver and spleen, organs were cut into small pieces and placed for 30 min at 37 °C at 200 RPM in 20 mL digestion media containing 4 μL of Collagenase IV Stock Solution (Sigma-Aldrich, Cat #C5138) per mL digestion media and 0.5 μL of DNAse 1 Stock solution per mL digestion media where digestion was halted using 5% FCS PBS solution on ice. The digested samples were then transferred through a 100 μM mesh and centrifuged at 50 g for 3 min. The supernatant was transferred and centrifuged for $400 \times g$ for 7 min. Next, the

pellet was resuspended in RPMI, and lympholyte was carefully layered underneath this solution. This solution then underwent density-gradient centrifugation with Lympholyte at $850 \times g$ at room temperature (acceleration 9, deacceleration 0), where the middle layer was kept for further analysis.

### Flow cytometry analysis

Cells were surface-labeled with the indicated antibodies for 30 min at 4° C. The following antibodies were used (dilution for each per manufacture instructions) for flow cytometry analysis: anti-CD62L-PerCP/Cy5.5 (BioLegend, Cat #104432, Clone: MEL-14, dilution 1:200), anti-CD44-APC/Cy7 (BioLegend, Cat #103028, Clone: IM7, dilution 1:200), anti-CD69-BV650 (BioLegend, Cat #104541, Clone:H1.2F3, dilution 1:100), anti-NK1.1-BV510 (BioLegend, Cat #108738, Clone: PK136, dilution 1:200), anti-B220-Alexa Fluor 700 (BioLegend, Cat #103232, Clone: RA3-6B2, dilution 1:400), anti-CD3-Alexa Fluor 594 (BioLegend, Cat #100240, Clone: 17A2, dilution 1:200), anti-CD3-Alexa Fluor 700 (BioLegend, Cat #100216, Clone: 17A2, dilution 1:200), anti-CD4-BV605 (BioLegend, Cat #100451, Clone: GK1.5, dilution 1:200), anti-CD4-BV510 (BioLegend, Cat #100449, Clone: GK1.5, dilution 1:200), anti-CD8-BV786 (BD Biosciences, Cat #563332, Clone: 53-6.7, dilution 1:200), anti-CD39-APC (BioLegend, Cat #143809, Clone: Duha59, dilution 1:100), anti-TIM-3-BV421 (BioLegend, Cat #119723, Clone: RMT3-23, dilution 1:100), anti-TIM-3-BV605 (BioLegend, Cat #119721, Clone: RMT3-23, dilution 1:100), anti-TIM-3-APC (BioLegend, Cat #119706, Clone: RMT3-23, dilution 1:100), anti-PD-1-FITC (BioLegend, Cat #135214, Clone: 29 F.1A12, dilution 1:100), anti-Ly-6G-BV650 (BioLegend, Cat #127641, Clone: 1A8, dilution 1:100), anti-Ly-6C-FITC (BioLegend, Cat #128006, Clone: HK1.4, dilution 1:100), anti-CD19-PErCp-Cy5.5 (BioLegend, Cat #152406, Clone: 1D3/CD19, dilution 1:200), anti-CD25-PerCp-Cy5.5 (BioLegend, Cat #101911, Clone: 3C7, dilution 1:200), anti-TCR-b -PE/Cy7 (BioLegend, Cat #109222, Clone: H57-597, dilution 1:300), anti-LAG-3-BUV496 (BD Biosciences, Cat #750027, Clone: C9B7W, dilution 1:100), anti-CD45-PerCp-Cy5.5 (BioLegend, Cat #157208, Clone: S18009F, dilution 1:100), anti-F4/80-Alexa Fluor 700 (BioLegend, Cat# 123130, Clone: BM8; dilution 1:400), anti-F4/80-FITC (BioLegend, Cat #123108, Clone: BM8, dilution 1:100), anti-F4/80-APC (Invitrogen, Cat #17-4801-80, Clone: BM8), anti-F4/80-PE (BioLegend, Cat #111704, Clone: W20065D, dilution 1:100) anti-CD11b-Alexa Fluor 700 (BioLegend, Cat #101222, Clone: M1/70, dilution 1:400), anti-CD11b-APC/Cy7 (BioLegend, Cat #101226, Clone: M1/70, dilution 1:200), anti-CD11b- BUV496 (BD Biosciences, Cat #749864, Clone: M1/70, dilution 1:200), anti-CD68- PerCp-Cy5.5 (BioLegend, Cat #137010, Clone: FA-11, dilution 1:200), anti-CD80-BV650 (BioLegend, Cat #104732, Clone: 16-10A1, dilution 1:100), anti-CD86-BV605 (BioLegend, Cat #105037, Clone: GL-1, dilution 1:100), anti-CD163-APC/Cy7 (BioLegend, Cat #155324, Clone: S150491, dilution 1:100), anti-CD206-AF594 (BioLegend, Cat #141726, Clone: C068C2, dilution 1:100), anti-CD274(PD-L1)-BV786 (BD Biosciences, Cat #741014, Clone: MIH5, dilution 1:100), and anti-CD183(CXCR3)-BV510 (BioLegend, Cat #126527, Clone: CXCR3-173, dilution 1:100). The following tetramers (dilution 1:200), from the NIH tetramer core facility in collaboration with Emory University, were used: OT-1 tetramer on APC, OT-1 tetramer on PE, AH1 tetramer on APC, AH1 tetramer on PE, TRP2 tetramer on APC, and TRP2 tetramer on PE. The following antibodies were used for intracellular cytokine and transcription factor staining using the BD Cytofix/Cytoperm™ Fixation/Permeabilization Kit (BD Biosciences, Cat #554714) or the BD Pharmingen™ Transcription Factor Buffer Set (BD Biosciences, Cat #562574) following the manufacturer's recommendation: anti-Granzyme B-APC (BioLegend, Cat# 396408, Clone: QA18A28, dilution 1:100), anti-Granzyme B-FITC (BioLegend, Cat #515403, Clone: GB11, dilution 1:100), anti-IFN-y-BV650 (BioLegend, Cat #505832, Clone: XMG1.2, dilution 1:100), anti-OPN-PE (R&D Systems, Cat #IC808P, dilution 1:100), Osteopontin Rabbit PolyAb-CoraLite Plus 488 (Proteintech, Cat#CL48822952100UL, dilution

1:100), anti-CXCL9-APC (BioLegend, Cat #515606, Clone: MIG-2F5.5, dilution 1:100), anti-Fibronectin-1 Alexa Fluor 647 (BD biosciences, Cat #563098, Clone: 10/Fibronectin, dilution 1:200), anti-FoxP3-BV421 (BioLegend, Cat #126419, Clone: MF-14, dilution 1:100), anti- FoxP3-PerCp-Cy5.5 (BD Biosciences, Cat #563902, Clone: R16-715, dilution 1:100), anti- Ki-67-BV421(BioLegend, Cat #652411, Clone: 16A8, dilution 1:100), anti- Ki-67-BV421 (Invitrogen, Ref #364-5698-82, dilution 1:100), and anti-Ki-67-APC (BioLegend, Cat #652405, Clone: 16A8, dilution 1:100). Samples were run on a CytoFLEX LX flow cytometer (Beckman Colter CytoFLEX Flow Cytometer, RRID: SCR_019627) and data was analyzed using FlowJo software (FlowJo, RRID:SCR_008520) software.

### Antibody depletion
Depletion of CD8 + T cells were conducted by 100 µL intraperitoneal injection containing 10 µg of antibody per gram of mouse weight (InVivoMAb anti-mouse CD8α, CAT# BE0061, Clone 2.43, BioxCell). Isotype-matched IgG2b (InVivoMAb rat IgG2b isotype control, anti-keyhole limpet hemocyanin, CAT# BE0090,Clone LTF-2, BioxCell) was administered at 10 µg antibody/g mouse to IgG controls.

### scRNA-Seq Analyses
The following previously published human data sets were used for single cell RNA seq analysis: GSE164522 (primary CRC and corresponding liver metastasis)[24], dbGap phs003279.v1.p1 (HCC)[38], GSE245535 (bulk mRNA-seq)[48], and GSE267209 (pancreatic cancer and corresponding metastasis)[48]. Previously published data with metadata and annotations was placed onto Biowulf (NIH HPC Linux cluster), and downstream analyses was performed using the Seurat package (vs 4.3.0.1) under Rstudio (vs 2023.09.1 + 494)[24,48].

For the CRC metastasis data, the count matrix files were read using the fread() function, then converted into Seurat objects, and the associated published metadata was added using the AddMetaData() function. The original QC was verified, including removing low-quality cells with the following criteria: cells with features fewer than 300 or above 6000 genes and nCounts above 10000. Normalization was performed using the NormalizeData() function. The FindVariableFeatures() function was used to identify 2000 highly variables genes to use for downstream analysis. These features were scaled using the ScaleData function. The RunPCA function was the method used for linear dimensional reduction. The FindNeighbors and FindClusters functions were used to find appropriate clusters. UMAPs were created using the RunUMAP function. RunUMAP was adjusted through terms of n.neighbors and dims to distinguish clusters. Differential gene analysis was completed for all clusters, using FindAllMarkers() and FindMarkers().

For the pancreatic cancer dataset, the h5 files were read using the Read10X_h5() function. The following QC metrics were applied: cells with features above 3000 or few than 200, nCounts less than 10000, log10 genes per UMI above 0.82, and mitochondrial gene percentage below 10. Each cell was ascribed a cell cycle score using the CellCycleScoring() function with default settings, and a cell cycle difference was calculated using the S phase minus the G2M phase. These cell cycle differences as well as mitochondrial percentage, ribosomal percentage, read depth (nFeature and nCount) were regressed. Original UMAPs were verified using the FindNeighbors() function, RunUMAP function (n.neighbors set to 75 L, min.dist = 0.1 and metric "manhattan").

### Identification of tumor reactive T cells.
Gene set enrichment analysis was performed on 63,284 cells previously annotated as CD8 + T cells[13,70]. The widely utilized methodology used to determine tumor reactivity was taken from a previous publication, including discussion with the paper's authors[13]. Briefly, a list of genes defining tumor reactivity was obtained from a recent study[14]. This set of genes was stored as an array in a list where the gene names would correspond to the later row names of the data matrix. The RNA data were retrieved using the GetAssayData() function, which were then converted to a matrix. The matrix was then used to calculate the gene expression for the original vector containing gene names (in this case, row names for the matrix). These scores were calculated for each individual column where each column is an individual cell. Once these values were stored in the original Seurat object's metadata, each patient was separated using a for loop, and the scores were z-scaled for cross-signature comparisons across both patients and clusters. Cells with z-scaled scores above 2 standard deviations were defined as potentially tumor reactive and retained as true in a TASBinary metadata column for further analysis. After creating the combined Seurat object of the CD8 + T cells, the values were plotted for supplementary analysis using RidgePlot() with the group.by set to patient and cluster, separately, and FeaturePlot() with the scale_color_gradientn with limits set to c(− 3, 3)). Furthermore, as advised by the original publication's authors, we strengthen the validity of our findings by repeating the analysis on 500 random genes. First, all gene names were retrieved using the PBMC CD8 + T cell Seurat object count's dimnames (dimension names) and stored as an array. Next, the array of genes was randomly selected for 500 variables using the sample() function with n set to 500 and replace set to the default "False". For these genes/rownames, the same calculation per patient, z-scaled values were stored in the metadata and compared between patients.

### CellChat.
CellChat analysis was completed using the createCellChat() function, completed both on major and sub-cluster labeling[29]. The CellChat.human database was used. The identifyOverExpressedGenes(), identifyOverExpressedInteractions(), and computeCommunProb() where the population.size was set to TRUE. This allows appropriate mathematical compensation for the population size when computing cell-cell interactions. The computeCommunProbPathway and aggregateNet() function was then used. From the CellChat objects, senders and receivers were identified and subsetted. The vertex weight for visualization based on a group size Input parameter in each cluster. The sender and receiver interaction strength was visualized using netVisual_circle(), netVisual_chord_gene(), netVisual_aggregate(), differentialPathwaySubChord(), subsetCommunication(), extractGeneSubsetFromPair(), and netVisual_bubble() functions. The compareInteractions() function was used to compare the number of cell-cell interactions after merging the relevant CellChat objects. Heatmaps were created using netVisual_heatmap() to visualize differences in the number and strength of interactions of different patient groups. This analysis was also completed on our previously published HCC single-cell RNA sequencing data to analyze SPP1 pathways in HCC patients[38].

### Expression analysis.
The Seurat object was first subsetted for each patient using a for loop. The percent of cells that express a certain gene is calculated using Percent_Expressing() with default settings, except in cases where cluster annotation was needed (in these cases, group.by setting was set to the appropriate annotation level to determine percent expressing cells in each cluster)[71]. Further per patient analysis was completed in Prism using linear analysis for statistical significance (statistical section below).

### Monocle/trajectory analysis.
Monocole and Pseudotime analysis were completed using the Monocle3 package (version 1.3.4)[72]. Briefly, for all monocole analyses, the Seurat objects were converted using the as.cell_data_set() and estimate_size_factors() functions. Afterwards, for the macrophage monocole analysis, the seed was set using the set.seed() function for reproducibility. The plot_cells function was then used with show_trajectory_graph set to false and label_roots set to true. Now completing trajectory analysis, the learn_graph and order_cells functions were used with default settings. The graph_test function was then used to learn the gene order with neighbor_graph set to

"principal_Graph" and cores set to 16. For relevant portions of the mainstream analysis for subset analysis, the choose_graph_segments function was used with clear_cds set to false. At all times, the resolutions were set as close to the original resolution of the labeled clustering annotations as possible.

## CODEX Analyses

Previously acquired and reported CODEX images (n = 15) from human HCC fresh frozen specimens stained for 37 protein targets (https://doi.org/10.7937/bh0r-y074)[36] were reanalyzed using HALO Image analysis software (version 3.6 from Indica Labs available through the NCI HALO Image Analysis Resource. The tissues were stained with 37 antibodies (a combination of commercially available and custom antibodies), washed, and fixed according to the recommended protocol from Akoya Biosciences. The signals were revealed in multiple cycles on a Keyence microscope using a Nikon Plan Apo lambda 20X lens with a 0.75 numerical aperture. The acquired images were processed using CODEX processor version 1.7.0.6 without segmentation, and the stitched single signal images were converted to pyramidal tiff images using bftools (supported by CBIIT at CCR, NCI). The pyramidal images were stored on a CBIIT-supported cloud storage, and selected signals (20 markers) relevant to the study and DAPI were fused into a single composite image using HALO. aSMA-positive pre-fibrotic regions were manually annotated to generate aSMA + and aSMA- regions in the tumor, rim, and normal liver regions. For single cell analysis, HALO's HiPlex FL module 4.2.14 was used with built-in default AI nuclear segmentation that recognizes nuclei with high precision. The settings for nuclear recognition on DAPI signal were the following: nuclear contrast threshold 0.5, the minimum nuclear intensity 0.01, nuclear size 4–400 μm, nuclear segmentation aggressiveness 0.5. The cell size was set to 6–600 μm with a cytoplasm radius of 2 μm. Each marker was manually thresholded (gated) for positivity and 19 different phenotypes were defined based on "and positive", "and negative" criteria in HALO. Gating was performed for each signal and each tissue separately. The analysis to identify cells that are positive for different signals and phenotypes was run separately on aSMA+ and aSMA- regions, as well as normal, tumor, and rim annotation layers. In addition, multiple pairwise proximity analysis and density heatmap analysis was performed on the obtained single-cell level data[38].

The single cell level numerical data was generated based on cell segmentation and provides spatial feature tables that include marker intensities, cell coordinates, cell and nuclear size, call for signal positivity, cell phenotype positivity, tissue annotation layer, etc, similarly to count tables in scRNA sequencing. These CSV files containing single-cell data are used for further downstream analysis.

## Peritoneal macrophage isolation and purification

Peritoneal macrophage isolation has been previously described[35]. Briefly, 5 mL of cold PBS with 3% FCS was injected into the peritoneal cavity of mice with a 19 g needle. The filled peritoneum was gently massaged to displace cells into the liquid. The peritoneal fluid was then drained and collected in a 15 mL tube. This washing technique was repeated once more. An incision was made in the peritoneum, and a plastic transfer pipette was used to collect any remaining liquid. Macrophages were purified using anti-F4/80 microbeads (Miltenyi Biotec, Cat # 130-110-443) and sorted using an AutoMACS Pro (Miltenyi Biotec). Cells were counted, and the amount of microbeads suggested in the manufacturer's instructions was doubled. Validation of macrophage purity was confirmed using flow cytometry as described above.

## Macrophages hypoxia induction

Macrophages were isolated as described above. The purified macrophages were plated on low-adherence plates and placed under normoxic conditions (5% CO2 and 37 °C). Similarly, the macrophages for hypoxia were plated on low-adherence plates and placed under hypoxic conditions (0.5% O2) for 14 h in the Whitley H35 Hypoxystation (Don Whitley Scientific).

## RNA isolation and RT-qPCR

**In vitro macrophage.** Intraperitoneal macrophages were isolated and put into normoxic conditions (5% $CO_2$, 37 °C) and hypoxic conditions (5% $O_2$, 95% $N_2$, 37 °C) and harvested for RT-qPCR. Cells were first collected for RNA isolation using the RNeasy Mini Kit (Qiagen, Cat #74106)[73]. cDNA was synthesized using the iScript cDNA Synthesis Kit (Bio-Rad, Cat #1708891). The following primers were used for quantitative PCR analysis: Gapdh (F: AAGTGGTGATGGGCTTCCC, R: GGCAAATTCAACGGCACAGT) and Spp1(F: TCTCCTTGCGCCACAGAATG, R: TGTGGTCATGGCTTTCATTGGA).

**In vivo mouse liver tissues.** RT-qPCR analysis was performed using SsoAdvanced Universal SYBR Green Supermix (Bio-Rad, Cat #1725271) from mouse liver tissues using the following primers, HIF1-a(F: CATAAAGTCTGCAACATGGAAGGT, R: ATTTGATGGGTGAGGAATGGGTT), Gapdh (F: AAGTGGTGATGGGCTTCCC, R: GGCAAATTCAACGGCACAGT), COL4a1(F: GGTGTCAGCAATTAGGCAGGTCAAG, R: ACTCCACGCAGAGCAGAAGCAAGAA), Spp1(F: TCTCCTTGCGCCACAGAATG, R: TGTGGTCATGGCTTTCATTGGA), and FN1(F: ATCTGGACCCCTCCT, R: GCCCAGTGATTTCAG) based on established PCR protocols[74–77]. Gapdh provided an endogenous control for genes of interest and were used for normalization. ddCT calculations provided relative expression to control samples. RNA was first extracted using RNeasy Mini Kit (Qiagen, Cat #74106) and checked the concentration and quality using Nanodrop (ThermoFisher Scientific, Cat #ND-2000). cDNA was synthesized using the iScript cDNA Synthesis Kit (Bio-Rad, Cat #1708891) and a PCR thermal cycler (Bio-Rad, Cat #1852148). qPCR was performed on an Applied Biosystems 7500 machine, with SsoAdvanced Universal SYBR Green Supermix (Bio-Rad, Cat #1725271), in 10 μL reaction volumes using 2 μl template DNA.

**In vitro CD8 + T cell osteopontin co-culture.** For T cell activation and co-culture, 96-well flat-bottom plates were coated overnight with anti-CD3 (BioXCell, Cat #BE0001-1) antibody, anti-CD28 antibody (BioXCell, Cat #BE0015-1), mouse recombinant osteopontin protein (Biolegend, Cat #763606), and anti-IgG (BioXCell, Cat #BE0090) antibody, as indicated by the incubation conditions. CD8 T cell isolation was completed according to manufacturer specifications (Miltenyi Biotec, Cat #130-104-075)[78]. and incubated with 10 μg/mL anti-CD44 antibody (BioXCell, Cat #BE0039) and plated on pre-coated plates[79]. After the appropriate incubation time, cells were either stained for flow cytometry or harvested for Nanostring analysis.

## Nanostring analysis

RNA of the cells was isolated using the Total RNA Purification Kit (STEMCELL Technologies Cat# 79040)[80]. RNA was hybridized to the NanoString Immune Exhaustion panel and read on the NanoString GEN2 nCounter Analysis System (NanoString). The expression of 785 immune-related genes and 12 housekeeping genes were assessed. The nSolver 4.0 software was used to normalize expression values using housekeeping genes following the manufacturer's recommendations. Samples were grouped into IgG, Spp1, and Spp1 + aCD44. GSEA and hierarchical clustering analysis were carried out on normalized data using manufacturer recommendations of the advanced analysis toolset within nSolver.

## Statistics & reproducibility

Sample sizes for animal studies were guided by previous studies with similar or identical tumor models (no statistical method was used to predetermine sample size)[38,68,69,81–83]. No data were excluded from the analyses. The majority of experiments were repeated at least twice to obtain robust data for the indicated statistical analyses.

For all readouts, examiners were blinded. The experiments (mice per group) were randomized. GraphPad Prism 9 and 10 (GraphPad Prism, RRID:SCR_002798) was utilized for statistical and visual output. Human data analyses were performed using R (vs. 4.2.1) on RStudio (vs. 2022.070.01). *P*-value < 0.05 indicates statistical significance.

### Reporting summary

Further information on research design is available in the Nature Portfolio Reporting Summary linked to this article.

## Data availability

The human single-cell sequencing data from HCC patients are deposited through dbGaP (accession number: phs003279.v1.p1): https:// www.ncbi.nlm.nih.gov/projects/gap/cgi-bin/study.cgi?study_id= phs003279.v1.p1 (HCC)[38]. The raw CODEX images are hosted at The Cancer Imaging Archives (TCIA) under https://doi.org/10.7937/bh0r-y074. Regarding publicly available datasets not generated by our lab, the following accession numbers provide access to both raw and processed data: GSE164522 (primary CRC and corresponding liver metastasis; https://www.ncbi.nlm.nih.gov/geo/query/acc.cgi?acc= GSE164522)[24], GSE245535 (bulk mRNA-seq; https://www.ncbi.nlm.nih. gov/geo/query/acc.cgi?acc=GSE245535)[48] and GSE267209 (pancreatic cancer and corresponding metastasis; https://www.ncbi.nlm.nih.gov/ geo/query/acc.cgi?acc=GSE267209)[48]. Source data are provided in this paper.

## Code availability

The trajectory and relevant code can be found in the repository (https://github.com/rajuvee/TASrev2)[85].

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

## Acknowledgements

We are grateful to Dr. Benjamin Ruf, Dr. Sepideh Babaei, Sophie Wang, Dr. Jay Berzofsky, Dr. Siobhan Corbett and Dr. Changqing Xie. This work utilized the computational resources of the NIH HPC Biowulf cluster (https://hpc.nih.gov). Nanostring nCounter digital gene expression was performed by the CCR Genomics Core at the National Cancer Institute/NIH (Bethesda, MD). We acknowledge the use of Biorender[84]. This research was supported [in part] by the Intramural Research Program of the NIH. RT and FJR-M receives funding through the NIH Medical Research Scholars Program. Research support was provided by the NIH Medical Research Scholars Program, a public-private partnership supported jointly by the NIH and contributions to the Foundation for the NIH from the American Association for Dental Research and the Colgate-Palmolive Company. TFG laboratory funding provided by the Intramural Research Program of the NIH, NCI (ZIA BC011345, ZO1 BC010870; TFG). TFG was support by the NCI FLEX award (TFG).

## Author contributions

R.T., P.H., C.M., F.K., and T.F.G. conceived and planned the experiments. R.T., P.H., X.B.Z., X.W., M.S., A.N., S.G., F.R., and C.M. carried out the experiments. R.T. planned and carried out the computational analysis of RNA-sequencing and spatial proteomic datasets. R.T., P.H., and N.K. curated the CODEX dataset, including annotations of cell types and regions. R.T., P.H., D.S., M.A., C.M., F.K., and T.F.G. contributed to the interpretation of the results. R.T., F.K., and T.F.G. wrote the manuscript. F.K. and T.F.G. supervised all aspects of the manuscript. All authors provided critical feedback and helped shape the research, analysis, and manuscript.

## Funding

## Competing interests

The authors declare no competing interests.
