## [Transparent Peer Review file · Nature Communications]

SPP1+ Macrophages Cause Exhaustion of Tumor-Specific T cells in Liver Metastases

Corresponding Author: Dr Tim Greten

Version 0:

Reviewer comments:

Reviewer #1

(Remarks to the Author)

The manuscript titled A Paradoxical Tumor Antigen Specific Response in the Liver provides significant insights into tumor-reactive CD8+ T cells, focusing on their dysfunction in liver metastasis. The authors employ a comprehensive approach, combining scRNA-seq, proteomic data, and murine models to address a critical issue in cancer immunology.

Here are my review suggestions:

Specific comments:

1. Compared to the splenic injection or portal vein injection models, the direct intrahepatic injection model used in this article might have several potential drawbacks:

Lack of natural metastatic pathway: Splenic and portal vein injections better simulate the natural metastatic route. In contrast, direct intrahepatic injection bypasses this process, which may not fully reflect the biological behavior of cancer cells migrating through the circulatory system and settling in the liver.

Impact on the immune microenvironment: Injecting tumor cells directly into the liver might alter the hepatic microenvironment differently from the gradual changes that occur in a natural metastatic process. This model may not accurately capture the immune microenvironment alterations that happen during the progression of liver metastasis.

In summary, while direct intrahepatic injection is a fast and reliable method to establish liver tumors, it may have limitations compared to splenic or portal vein injections in terms of simulating natural metastatic pathways, systemic evaluation, and immune microenvironment dynamics.

Minor comments:

1. "On day 14, the frequency of TAS CD8+ T cells was higher in the intrahepatic TIL compared to subcutaneous TIL ($p = 0.0021$), whereas no significant differences in tumor weights between intrahepatic and subcutaneous CT26 tumors were found at this time point, ruling out the contribution of tumor burden ($p = 0.3792$) (Figure 1B)." sections like this involving p-values should specify the statistical methods. This will help readers understand the rationale behind the choice of statistical tests and the reliability of the significance of the data.

2. "whereas no significant differences in tumor weights between intrahepatic and subcutaneous CT26 tumors were found at this time point, ruling out the contribution of tumor burden ($p = 0.3792$) (Figure 1B)." here should be Figure 1D, Please ensure the consistency between the figures and the text in the manuscript.

3. The manuscript mainly includes bar plots. It may be beneficial to diversify the types of figures to make the data presentation more varied and engaging. Additionally, the overall aesthetics of the figures could be improved. Some of the figures are difficult to interpret, simplifying or clarifying these figures would make the results more accessible to readers.

4. It is not clearly stated which data were newly generated by the authors and which were previously published by others.

5. "We used single-cell Gene Set Enrichment Analysis (scGSEA) with more than 100 gene signatures to identify potentially tumor-reactive (pTRT) human CD8+ T cells (Figure 3A). We applied highly stringent cutoffs for the identification of pTRT cells to facilitate the identification of biologically valid mechanisms. CD8+ T cells (1,310 cells) that met this cutoff were labeled as pTRT cells" Please provide a detailed description in the methods section of how pTRT cells were defined.

Reviewer #2

(Remarks to the Author)

This study investigated the tumor immune responses in liver by examining the dynamic cell population changes in multiple mouse models. The scRNA-seq and CODEX analysis get interesting observations for the tumor-specific CD8+ T cells, but a few concerns about the method details need to be resolved:

1. Some of the correlations in Figs 2behknq are misleading. For example, in Fig. 2B, the points are located at two extremes.
2. One of the most important steps in this study is to identify the pTRT sub-populations. The provided description is insufficient to evaluate the reliability of the method.
3. The computational analysis of scRNA-seq is somehow tricky. More details should be provided, for example, how to define the number of clusters and how to conduct the trajectory inference of macrophages/monocytes, scGSEA settings, etc. The codes and annotations are better provided via github or other public site.
4. More comparisons with clinical datasets should be conducted, especially for the major observations. The cell line injected mouse models are highly different with the primary human tumors.
5. The key word "paradoxical" in the title is not strongly indicated by the data and experiments.

Reviewer #3

(Remarks to the Author)

The manuscript by Trehan et al. provides an in-depth exploration of the status and function of tumor-reactive CD8+ T cells in the liver tumor microenvironment, with a particular focus on the role of SPP1+ macrophages and their relationship with the profibrotic environment.

The authors first identified that high frequency of TAS CD8+ T cells in intrahepatic tumors compared to subcutaneous tumors. Then, the exhausted and dysfunctional TAS CD8+ T cells was found in mice with liver tumors. Then, a high frequency of pTRT CD8+ T cells was also observed in patients with hepatic metastasis. Furthermore, the authors demonstrated that the relationship and interactions between SPP1+ macrophages and TAS CD8+ T cell exhaustion and dysfunction. Finally, they identified that local and distal tumor signaling correlated to the profibrotic polarization and enrichment of an IC population in the liver.

1. Regarding the results in "SPP1+ macrophages interact with pTRT cells in humans," the authors should consider the critical factor of the timing of liver metastasis. First, the ability of CD8+ T cells to correctly recognize tumor cells depends on the activity and efficiency of antigen-presenting cells. As one of the most important antigen-presenting cells, macrophages' interactions with T cells will inevitably vary at different time points of tumor development. Therefore, differences in CellChat analysis between the primary site and the liver metastatic site are expected. The authors should clarify the time points of liver metastasis for the selected samples and explain the rationale behind them to enhance the persuasiveness of the results.
2. The manuscript mentions that SPP1+ macrophages induce CD8+ T cell exhaustion through SPP1-CD44 interactions. However, the specific signaling pathways and downstream molecular mechanisms need further elucidation. It will be better to provide more experimental data, particularly at the level of signaling pathways, to support this conclusion.
3. In the fourth section, "Macrophage and CD8+ T cell interactions are enriched in α SMA+ environments," the authors should explain why the "B16F10" cell line was used. It is well known that melanoma, unlike other malignant tumors, is a highly immunogenic tumor with abundant T cell infiltration and is more sensitive to immunotherapy. However, for most malignant tumors, immunogenicity is generally poor. The authors should provide more substantial evidence to demonstrate that this phenomenon is not specific to melanoma.
4. In the last part of the manuscript, the authors analyze single-cell data from pancreatic cancer patients. It is worth noting that extensive fibrosis is one of the most important characteristics of pancreatic cancer. The authors should clarify the reason for choosing pancreatic cancer data for analysis in this context.

Minor points:

1. The experimental schematic diagram in Figure 1A is a summary of Figure 1 B-D. Similarly, the experimental schematic diagram in Figure 1E is a summary of Figure 1 F-G. However, the description of "harvest" in the final stage of the diagram is so simple. It is better to describe the steps after collection or the collected items in detail.
2. The legend should be described in detail in Figure 1B (such as IH, SQ, TF) and Figure 2. Moreover, the P value should be showed.
3. The picture "TF" was not found in Figure 1 C-D, and the number should be consistent with each groups in Figure 1 C, D, F and Figure 2 T, U. The weights between IH and SQ at 14days in Figure 1 D was not statistically analyzed. The bar should be displayed in the diagram.
4. The P-value represented by the asterisk should be described more clearly in legend.
5. The figure does not match the description in the result section "Phenotype and function of TAS CD8+ T cells"(Figure1 G and 2A).
6. The software used to calculate statistical significance should be clearly described in Statistics section.
7. The methods section should be described in more detail.

Version 1:

Reviewer comments:

Reviewer #1

(Remarks to the Author)

The authors have revised the manuscript to address all the comments raised previously. I have no further comments.

Reviewer #2

(Remarks to the Author)

1. The authors did not directly response to the question about whether the correlation(s) is misleading if the data points are located at two extremes. It is a common mistake for correlation analysis.

2. The validations can be implemented on un-paired clinical samples.

Reviewer #3

(Remarks to the Author)

The authors have addressed all my concerns.

Version 2:

Reviewer comments:

Reviewer #2

(Remarks to the Author)

Cannot the authors frankly admit the mistake for the correlation analysis?

The removal of the corresponding sub-figures is just an action taken solely to resolve this issue.

I have no other comments.

Point-by-point reply

Manuscript #: **NCOMMS-24-52124-T**

(Title) **“SPP1+ Macrophages Cause Exhaustion of Tumor-Specific T cells in Liver Metastases”**

GENERAL COMMENTS

We thank the reviewers and the editorial team for their many insightful and constructive suggestions. In response, we have conducted additional experiments, performed further analyses, added new *in vitro/in vivo* data, and reorganized significant portions of the manuscript and figures.

1. We conducted experiments in mice after splenic injection of tumor cells.
2. We conducted *ex vivo/in vitro* experiments with SPP1 protein to study downstream pathways.

In summary, new results/analyses added to this revision clarified the mechanism and revealed putative signaling intermediates and mechanisms of metastatic tumor burden.

To assist editors & reviewers, comments from the editors and reviewers are reported in *italic* and replies in regular type. Figures labeled with a capital letter refer to figures presented in the point-by-point reply. We also provide the # of the figure in the revised manuscript. A copy of manuscript changes presented in this point-by-point reply can be identified by a light grey background. In the revision, **Figures 1, 2, 3, 4, 5, 7, 8, and 9 as well as supplemental figures 3, 8 and 9** were revised/edited. The following figures contain significant new data:

Main Figure: Fig. 4 and 9

Supplementary Figure: 8

Editor:

In particular, we would expect your revision to justify the use of the intrahepatic model, as requested by Reviewer #1, and to provide further details of the analysis throughout, as suggested by Reviewers #1 and #2. We would also expect your revision to include further experimental data for the mechanism, as suggested by Reviewer #3.

Response:

We would like to thank the editor for these helpful comments. We conducted additional studies using a different tumor model and injected tumor cells into the spleen to seed liver metastasis. We also conducted additional studies to better understand the mechanism as requested by reviewer #3.

REVIEWER #1:

The manuscript titled *A Paradoxical Tumor Antigen Specific Response in the Liver* provides significant insights into tumor-reactive CD8+ T cells, focusing on their dysfunction in liver metastasis. The authors employ a comprehensive approach, combining scRNA-seq, proteomic data, and murine models to address a critical issue in cancer immunology.

Here are my review suggestions:

Specific comments:

1. Compared to the splenic injection or portal vein injection models, the direct intrahepatic injection model used in this article might have several potential drawbacks:

Lack of natural metastatic pathway: Splenic and portal vein injections better simulate the natural metastatic route. In contrast, direct intrahepatic injection bypasses this process, which may not fully reflect the biological behavior of cancer cells migrating through the circulatory system and settling in the liver.

Impact on the immune microenvironment: Injecting tumor cells directly into the liver might alter the hepatic microenvironment differently from the gradual changes that occur in a natural metastatic process. This model may not accurately capture the immune microenvironment alterations that happen during the progression of liver metastasis.

In summary, while direct intrahepatic injection is a fast and reliable method to establish liver tumors, it may have limitations compared to splenic or portal vein injections in terms of simulating natural metastatic pathways, systemic evaluation, and immune microenvironment dynamics.

Response:

We thank the reviewer for this important comment. We conducted additional studies using the intra-splenic injection model. We injected RIL-175 cells intra-splenic in Spp1 knockout (KO) and wild-type (WT) mice. Mice were sacrificed on day 18 post-injection, and liver tumor burden was measured (Figure A1). As expected Spp1 KO mice had significantly lower tumor burden in the liver than wildtype mice (Figures A2 and A3) supporting our earlier results.

[REDACTED]

Figure A: Related to Figure 4. **1)** Experimental schematic for 2-3. RIL-175 tumor burden based on the presence of tumor **(2)** and number of nodules **(3)** in Spp1 KO (n=5, 4 male & 1 female) compared to WT mice (n=7; 3 male & 4 female). Data (2-3) was analyzed using an unpaired t-test.

Minor comments:

1. "On day 14, the frequency of TAS CD8+ T cells was higher in the intrahepatic TIL compared to subcutaneous TIL ($p = 0.0021$), whereas no significant differences in tumor weights between intrahepatic and subcutaneous CT26 tumors were found at this time point, ruling out the contribution of tumor burden ($p = 0.3792$) (Figure 1B)." sections like this involving p-values should specify the statistical methods. This will help readers understand the rationale behind the choice of statistical tests and the reliability of the significance of the data.

Response:

We agree with the reviewer's statement that a statistical test should accompany these p values. We report the statistical test for every analysis in the corresponding figure legends.

2. *"whereas no significant differences in tumor weights between intrahepatic and subcutaneous CT26 tumors were found at this time point, ruling out the contribution of tumor burden ($p = 0.3792$) (Figure 1B)." here should be Figure 1D, Please ensure the consistency between the figures and the text in the manuscript.*

Response:

Thank you very much for pointing this out. The labels of these figures were switched inadvertently when creating the final figures. We reviewed the entire manuscript to ensure that all figure labels are correct in the revised manuscript.

3. *The manuscript mainly includes bar plots. It may be beneficial to diversify the types of figures to make the data presentation more varied and engaging. Additionally, the overall aesthetics of the figures could be improved. Some of the figures are difficult to interpret, simplifying or clarifying these figures would make the results more accessible to readers.*

Response:

We thank the reviewer for the suggestion. We have changed the following main figures: 1,2,3,4,5,7, 8 and 9, as well as supplemental figures 3 and 9.

4. *It is not clearly stated which data were newly generated by the authors and which were previously published by others.*

Response:

We thank the reviewer for this suggestion, which points to the numerous data sets used in this study. We modified the text to include a data sources section in the methods so that the information is clearly provided. Additionally, we state the following in the main text,

- For Figures 3 and 6, we have stated that the CRC metastasis data set is "publicly available processed and annotated".¹
- For Figure 5, for our generated CODEX and scRNA-seq data, we have stated: "We recently performed a spatial co-detection by indexing (CODEX) imaging analysis as well as scRNA-seq analysis from liver tumors and adjacent tissue of HCC patients.⁴⁴ We repeated scGSEA on this HCC scRNA-seq data set with the same approach as our previous scRNA-seq analysis to define pTRT CD8+ T cells in these HCC patients (Figure 3A)."
- For Figure 7, the pancreatic cancer data set, we have stated "recently published scRNA-seq data of patients with pancreatic adenocarcinoma (Figure 7T)".²

5 *"We used single-cell Gene Set Enrichment Analysis (scGSEA) with more than 100 gene signatures to identify potentially tumor-reactive (pTRT) human CD8+ T cells (Figure 3A). We applied highly stringent cutoffs for the identification of pTRT cells to facilitate the identification of biologically valid mechanisms. CD8+ T cells (1,310 cells) that met this cutoff were labeled as pTRT cells" Please provide a detailed description in the methods section of how pTRT cells were defined.*

Response:

-We thank the reviewer for this comment. We have modified the text of the methods and main figure accordingly.

REVIEWER #2:

This study investigated the tumor immune responses in liver by examining the dynamic cell population changes in multiple mouse models. The scRNA-seq and CODEX analysis get

interesting observations for the tumor-specific CD8+ T cells, but a few concerns about the method details need to be resolved:

1. Some of the correlations in Figs 2behknq are misleading. For example, in Fig. 2B, the points are located at two extremes.

Response:

We appreciate the reviewer's feedback. We respectfully disagree that the figures are misleading. Comparative analysis of CD39 and AH1-specific T cells in the liver clearly demonstrates a correlation, which was not the case when we compared Gzmb and AH-1 specific T cells. In addition, we have moved some of the data to the Supplementary Figures to make it easier to understand the main message of the figure.

2. One of the most important steps in this study is to identify the pTRT sub-populations. The provided description is insufficient to evaluate the reliability of the method.

Response:

-We thank the reviewer for this comment. We have modified the text of the methods and main figure accordingly.

3. The computational analysis of scRNA-seq is somehow tricky. More details should be provided, for example, how to define the number of clusters and how to conduct the trajectory inference of macrophages/monocytes, scGSEA settings, etc. The codes and annotations are better provided via github or other public site.

Response:

-We thank the reviewer for this comment. We have modified the text of the methods and updated the figure to include more details accordingly. We also have uploaded our relevant code detailed in this paper to the TAS repository (<https://github.com/rajuvee/TASrev2>).

4. More comparisons with clinical datasets should be conducted, especially for the major observations. The cell line injected mouse models are highly different with the primary human tumors.

Response:

We thank the reviewer for this suggestion. We would like to bring to the reviewer's attention that we have used three different human RNA-seq datasets (a primary CRC with metastasis, HCC and PDAC) and a human CODEX data set throughout this study. Despite extensive literature search, we could not find other paired primary tumor and liver metastasis RNA-seq data sets.

5. The key word "paradoxical" in the title is not strongly indicated by the data and experiments.

Response:

We thank the reviewer for this comment. We modified the title accordingly.

REVIEWER #3:

The manuscript by Trehan et al. provides an in-depth exploration of the status and function of tumor-reactive CD8+ T cells in the liver tumor microenvironment, with a particular focus on the role of SPP1+ macrophages and their relationship with the profibrotic environment. The authors first identified that high frequency of TAS CD8+ T cells in intrahepatic tumors compared to subcutaneous tumors. Then, the exhausted and dysfunctional TAS CD8+ T cells was found in mice with liver tumors. Then, a high frequency of pTRT CD8+ T cells was also observed in patients with hepatic metastasis. Furthermore, the authors demonstrated that the relationship and interactions between SPP1+ macrophages and TAS CD8+ T cell exhaustion and

dysfunction. Finally, they identified that local and distal tumor signaling correlated to the profibrotic polarization and enrichment of an IC population in the liver.

1. Regarding the results in “SPP1+ macrophages interact with pTRT cells in humans,” the authors should consider the critical factor of the timing of liver metastasis. First, the ability of CD8+ T cells to correctly recognize tumor cells depends on the activity and efficiency of antigen-presenting cells. As one of the most important antigen-presenting cells, macrophages' interactions with T cells will inevitably vary at different time points of tumor development. Therefore, differences in CellChat analysis between the primary site and the liver metastatic site are expected. The authors should clarify the time points of liver metastasis for the selected samples and explain the rationale behind them to enhance the persuasiveness of the results.

Response:

Thank you for the comment. We appreciate that during cancer initiation and progression, macrophage phenotype and function vary drastically.³ Tumor samples from CRC patients were derived from patients with synchronous metastasis. Metastasis and primary tumors were collected at the same time point during a combined surgical resection. However, no published information from this dataset is provided regarding tumor burden in the liver compared to the colon, or time to development from primary lesion to liver metastasis (if even known at all, as some patients present with synchronous metastases or are worked up at multiple centers leading to loss of exactly timed progression data). Therefore, it would not be possible to determine the time between primary CRC tumor formation and liver metastasis, nor the age of the metastasis for these synchronous lesions.

For our mouse studies, all tumors were implanted on the same day excluding an effect of time on anti-tumor immune responses.

2. The manuscript mentions that SPP1+ macrophages induce CD8+ T cell exhaustion through SPP1-CD44 interactions. However, the specific signaling pathways and downstream molecular mechanisms need further elucidation. It will be better to provide more experimental data, particularly at the level of signaling pathways, to support this conclusion.

Response:

We conducted additional experiments to answer this important question. To address this, we conducted additional experiments to further investigate T cell exhaustion. CD8+ T cells were isolated (>98% purity) using a negative selection method from murine splenic lymphocytes and stimulated with plate-bound anti-C3 and anti-CD38 antibodies. Cells were incubated in T cell media for 24 hours under the following three experimental conditions: (1) IgG control, (2) plate-bound recombinant Spp1, or (3) recombinant Spp1 plus soluble anti-CD44 antibody (Figures B1 and B2).^{4,5} Cells were further analyzed using both flow cytometry and the Nanostring nCounter® Immune Exhaustion Panel to determine both T cell exhaustion and possible downstream pathways.

Following incubation, T cell exhaustion was assessed using Nanostring's T cell gene exhaustion score (Figure B3) and CD39+ protein expression (Figure B4). Both T cell exhaustion scores and CD39 expression were upregulated when stimulated with Spp1 protein, and this effect was reversed through the addition of anti-CD44. Furthermore, downstream advanced analysis of the immune exhaustion panel included in the Nanostring panel revealed key regulators of the exhaustion mediation. Specifically, downstream JAKSTAT, MAPK, NF-kB and TLR signaling pathways were found to increase upon stimulation with Spp1 (Figures B5 and B6). Interestingly, upregulation of these key pathways mediating T cell exhaustion was reversed upon blocking Spp1-CD44 interactions through soluble anti-CD44 antibody (Figures B5 and B6).

[REDACTED]

Figure B: Related to Figure 4. **1)** Experimental schematic for 2-6. **2)** Frequency of CD8+ T cells before purification compared to after MACS purification using a negative (no touch) isolation method. **3)** T cell exhaustion score in IgG

treated (n=2), Spp1 (n=2), and Spp1 with anti-CD44 Ab (n=2) treated CD8+ T cells. **4)** CD39 expression using flow cytometry by CD8+ T cells in IgG treated compared to Spp1 treated cells. **5)** JAKSTAT, MAPK, NF- κ B and TLR signaling scores as determined by Ncounter analysis. **6)** All pathway scores available for the Immune Exhaustion Panel by Nanostring.

3. In the fourth section, “Macrophage and CD8+ T cell interactions are enriched in α SMA+ environments,” the authors should explain why the “B16F10” cell line was used. It is well known that melanoma, unlike other malignant tumors, is a highly immunogenic tumor with abundant T cell infiltration and is more sensitive to immunotherapy. However, for most malignant tumors, immunogenicity is generally poor. The authors should provide more substantial evidence to demonstrate that this phenomenon is not specific to melanoma.

Response:

We believe there is a misunderstanding. All CODEX data is derived from human HCC samples.

4. In the last part of the manuscript, the authors analyze single-cell data from pancreatic cancer patients. It is worth noting that extensive fibrosis is one of the most important characteristics of pancreatic cancer. The authors should clarify the reason for choosing pancreatic cancer data for analysis in this context.

Response:

We appreciate the reviewer’s careful assessment of this scRNA-seq data as well as the fundamental drawbacks to this sequencing analysis. However, this is the only data set that exists for premetastatic liver sequencing with careful follow-up.

Minor points:

1. The experimental schematic diagram in Figure 1A is a summary of Figure 1 B-D. Similarly, the experimental schematic diagram in Figure 1E is a summary of Figure 1 F-G. However, the description of “harvest” in the final stage of the diagram is so simple. It is better to describe the steps after collection or the collected items in detail.

Response:

We appreciate the reviewer’s comments and have taken out the experimental schematic diagram, which is not necessary to understand the simple experimental set-up.

2. The legend should be described in detail in Figure 1B (such as IH, SQ, TF) and Figure 2. Moreover, the P value should be showed.

Response:

We appreciate the reviewer’s comment and have added P-values as well as information about the abbreviation to the figures.

3. The picture “TF” was not found in Figure 1 C-D, and the number should be consistent with each groups in Figure 1 C, D, F and Figure 2 T, U. The weights between IH and SQ at 14days in Figure 1 D was not statistically analyzed. The bar should be displayed in the diagram.

Response:

We routinely omit showing pictures of tumor-free livers if no cells were injected and only shows this if no tumors grew as a result of a specific intervention. Due to the complex surgical intervention, we do not always have the same number of mice in each group. We have added the statistical analysis to compare tumor weights on days 14 in previous Figure 1D (now Figure 1C). To address the second point, we have added in the nonsignificant (ns) bars option in Prism to clarify that analysis was completed on these groups as well. Additionally, we fixed the legend to state 11 samples for SQ α CD8 treatment.

4. The P-value represented by the asterisk should be described more clearly in legend.

Response:

P-value asterisk representations were added to figure legends.

5. The figure does not match the description in the result section “Phenotype and function of TAS CD8+ T cells”(Figure 1 G and 2A).

Response:

-Thank you very much for pointing this out, the labels of these figures were switched inadvertently when creating the final figures. This has been corrected in the updated version of the figure.

6. The software used to calculate statistical significance should be clearly described in Statistics section.

Response:

- We mention the following in the method’s section: “GraphPad Prism 9 and 10 (GraphPad Prism, RRID:SCR_002798) was utilized for statistical and visual output.”

7. The methods section should be described in more detail.

Response:

-We thank the reviewer for this comment. We have modified the text of the methods accordingly.

References:

1. Liu, Y. *et al.* Immune phenotypic linkage between colorectal cancer and liver metastasis.

Cancer Cell **40**, 424-437.e5 (2022).

2. Bojmar, L. *et al.* Multi-parametric atlas of the pre-metastatic liver for prediction of metastatic outcome in early-stage pancreatic cancer. *Nat Med* 1–11 (2024) doi:10.1038/s41591-024-

03075-7.

3. Qian, B.-Z. & Pollard, J. W. Macrophage Diversity Enhances Tumor Progression and Metastasis. *Cell* **141**, 39–51 (2010).

4. Klement, J. D. *et al.* An osteopontin/CD44 immune checkpoint controls CD8+ T cell activation and tumor immune evasion. *J Clin Invest* **128**, 5549–5560 (2018).

5. Norris, P. A. A. *et al.* Anti-inflammatory activity of CD44 antibodies in murine immune thrombocytopenia is mediated by Fcγ receptor inhibition. *Blood* **137**, 2114–2124 (2021).

Point-by-point reply

Manuscript #: **NCOMMS-24-52124A**

(Title) **“SPP1+ Macrophages Cause Exhaustion of Tumor-Specific T cells in Liver Metastases”**

GENERAL COMMENTS

We thank the reviewer and the editorial team for their insightful and constructive suggestions.

To assist editors & the reviewer, comments from the editors and the reviewer are reported in *italic* and replies in regular type. Figures labeled with a capital letter refer to figures presented in the point-by-point reply. We also provide the # of the figure in the revised manuscript. A copy of manuscript changes presented in this point-by-point reply can be identified by a light grey background. In the revision, **Figures 2 as well as supplemental figures 3** were revised/edited.

REVIEWER #1:

The authors have revised the manuscript to address all the comments raised previously. I have no further comments.

Response:

We thank the reviewer.

REVIEWER #2:

1. The authors did not directly response to the question about whether the correlation(s) is misleading if the data points are located at two extremes. It is a common mistake for correlation analysis.

Response:

We appreciate the reviewer's feedback. We removed Figure 2 and Supplemental Figure 3 correlations to address this comment. We believe that the removal of this data does not impact on the overall impact and conclusions drawn.

2. The validations can be implemented on un-paired clinical samples.

Response:

We thank the reviewer for this comment. Merging multiple separate sequencing runs while accounting for patient tumor variability requires extensive batch correction. In this manuscript, major murine findings have been validated using human datasets that incorporate spatial proteomics and multiple scRNA-seq datasets, with approximately half of the main figures presenting only clinical samples.

We would like to highlight that we have utilized three different human RNA-seq datasets (primary CRC with metastasis, HCC, and PDAC) as well as a human CODEX dataset throughout this study. Figures 1 and 2 show that mice exhibit a high frequency of exhausted, tumor-reactive CD8⁺ T cells, a pattern we also observe in the human scRNA-seq dataset in Figure 3. Similarly, Figure 4 demonstrates that in mice, SPP1 macrophages interact with exhausted, tumor-reactive CD8⁺ T cells, a finding that is consistent with our human scRNA-seq data in Figure 3. Moreover, Figures

6–8 primarily present human data. Finally, Figure 9 addresses a reviewer’s request for an intrasplenic model to mimic the human data observed in Figure 8.

To further address the reviewer’s suggestion, we integrated a fifth (!) clinical dataset, incorporating a large cross-validated scRNA-seq dataset containing 224,611 cells from human primary non-small cell lung cancer (NSCLC) tumors.¹ We initially completed brisk batch correction with the CRC dataset, including completing scaling/regression to match initial QC between datasets. Afterwards, we then separated the previously published T cell annotations for the NSCLC tumors, separating a total of 78,264 T cells from 89 patients. Clustering of the T cells revealed a total of 32,361 CD8 T cells from 5 clusters (Figure A1). Repeating the pTRT identification methodology (Figure 3A of our manuscript) revealed a total of 595 pTRT cells. The frequency of lung tumor pTRT CD8+ T cells in these 89 patients was significantly lower compared to hepatic metastasis (Figure A2). We have not included this figure in the manuscript since the data presented is not really different from what we already have in the manuscript, but this could be added upon editor’s request.

[REDACTED]

REVIEWER #3:

The authors have addressed all my concerns.

Response:

We thank the reviewer.

References:

1. Prazanowska, K. H. & Lim, S. B. An integrated single-cell transcriptomic dataset for non-small cell lung cancer. *Sci Data* **10**, 167 (2023).

Point-by-point reply

Manuscript #: **NCOMMS-24-52124B**

(Title) **“SPP1+ Macrophages Cause Exhaustion of Tumor-Specific T cells in Liver Metastases”**

REVIEWER #2:

1. Cannot the authors frankly admit the mistake for the correlation analysis?

The removal of the corresponding sub-figures is just an action taken solely to resolve this issue.

Response:

Reviewer 2 discusses the correlation of plots found in the original submission of Figure 2. Across the three comments by reviewer 2, the reviewer states that the points are located at two extremes (i.e., bimodal or multimodal datasets).

The p values shown in the figure are t-tests for non-zero slopes. If the data was bimodal, a bimodal distribution with values in both modes that are similar will provide non-significant p values. This can be seen with the Granzyme B and Ki-67 data. Thus, the reviewer's point was already invalidated as commented multiple times in our responses by proof of negative results with similar distribution of samples.

The point of these correlations was not to argue linearity or the shape of the distribution. Rather, we simply restated the same data found in the %AH1 graphs vs time as a scatter plot to show that at time points with higher TAS response also had higher expression of specific phenotypic markers. The reviewer, however, was highly concerned with the linearity for some reason even though we did not state it was necessarily linear.

However, we can further demonstrate this point by testing for the bimodal nature as the reviewer argues with Hartigan's dip test (Table 1).¹ We find that for almost every marker and AH1 frequency (every X and Y axis shown in the figure) the data is unimodal (p values < 0.05 indicate that the null hypothesis of unimodality cannot be rejected). Thus, the reviewer's original concern, about the bimodal nature of the data, is invalid.

We removed these correlations because the conclusion of the data could already be drawn from the previous figures and to avoid a fundamentally flawed debate when all other reviewer comments had already been addressed.

[REDACTED]

[REDACTED]

References:

1. dip.test function - RDocumentation.

<https://www.rdocumentation.org/packages/diptest/versions/0.77-1/topics/dip.test>.